# Aerosol-Induced Closure of Marine Cloud Cells: Enhanced Effects in the Presence of Precipitation

Matthew W. Christensen[1], Peng Wu[1], Adam C. Varble[1], Heng Xiao[1], and Jerome D. Fast[1]

[1]Atmospheric Science & Global Change Division, Pacific Northwest National Laboratory, Richland, WA 99354, Washington, USA

**Correspondence:** Matthew W. Christensen (matt.christensen@pnnl.gov)

**Abstract.** The Weather Research Forecasting (WRF) V4.3 model is configured within a Lagrangian framework to quantify the impact of aerosols on evolving cloud fields. Kilometer-scale simulations utilizing meteorological boundary conditions are based on 10 case study days offering diverse meteorology during the Aerosol and Cloud Experiments in the Eastern North Atlantic (ACE-ENA). Measurements from aircraft, the ground-based Atmosphere Radiation Measurement (ARM) site at Graciosa Island in the Azores, and A-Train and geostationary satellites are utilized for validation, demonstrating good agreement with the WRF-simulated cloud and aerosol properties. Higher aerosol concentration leads to suppressed drizzle and increased cloud water content in all case study days. These changes lead to larger radiative cooling rates at cloud top, enhanced vertical velocity variance, and increased vertical and horizontal wind speed near the base of the lower-tropospheric inversion. As a result, marine cloud cell area expands, narrowing the gap between shallow clouds and increasing cloud optical thickness, liquid water content, and the top-of-atmosphere outgoing shortwave flux. While similar aerosol effects are observed in lightly to non-raining clouds, they tend to be smaller by comparison. These simulations show a relationship between cloud cell area expansion and the radiative adjustments caused by liquid water path and cloud fraction changes. The adjustments are positive and scale as 74% and 51%, respectively, relative to the Twomey effect. While higher resolution large eddy simulations may provide improved representation of cloud-top mixing processes, these results emphasize the importance of addressing mesoscale cloud-state transitions in the quantification of aerosol radiative forcing that cannot be attained from traditional climate models.

## 1 Introduction

The surface temperature of Earth is kept cooler by the presence of low-level clouds, in particular stratocumulus. It has been estimated that a mere increase of about 4% in their global coverage would be enough to offset the radiative warming due to a doubling of atmospheric carbon dioxide (Randall et al., 1984). Aerosols, commonly emitted alongside greenhouse gases have the potential to decrease cloud droplet size and create more numerous droplets that effectively suppress precipitation and moisten the boundary layer (Albrecht, 1989). This process can increase the vertical and horizontal extents of clouds as well as their lifetime (Albrecht, 1989; Pincus and Baker, 1994; Bretherton et al., 2007; Christensen et al., 2020). However, an increase in aerosol concentration can also result in cloud desiccation due to enhanced cloud-top entrainment caused by more effective evaporation in polluted clouds (Ackerman et al., 2004; Small et al., 2009) or through reduced cloud droplet sedimentation

(Bretherton et al., 2007). These processes can even modify the cellular structure of clouds through changing cloud fraction (Rosenfeld et al., 2006). However, the strength and sign of the cloud radiative effect depends on a multitude of meteorological factors such as lower tropospheric stability and humidity, precipitation state (Chen et al., 2014), and the time-scale for which clouds have been polluted (Wang and Feingold, 2009). These complex relationships result in poor understanding and large uncertainty in estimates of rapid cloud adjustments to changes in aerosol concentration (Bellouin et al., 2020), the so-called aerosol-cloud lifetime effect (Albrecht, 1989). It is critical to quantify and resolve process-scale cloud physics impacting rapid adjustments in order to improve estimates of aerosol radiative forcing at global-scales (Seinfeld et al., 2016).

A preponderance of evidence linking aerosol and cloud radiative effects to the mesoscale structure of clouds has been growing in the literature over the past couple of decades (Rosenfeld et al., 2006; Wood, 2012; Christensen and Stephens, 2012; Eastman et al., 2021). Stratocumulus can exhibit cellular structures which appear closed or open with hexagonal-like or honeycomb shapes that organize on scales ranging from 10 – 50 km (Wood, 2005). The impact of aerosol on precipitation, as proposed by Rosenfeld et al. (2006), can reverse the direction of the wind flow through the vertical extent of the marine boundary layer, doubling cloud cover and converting cloud structure from open to closed cells. Eastman et al. (2021) observed that stronger surface winds and lower cloud droplet concentrations are typical prior to the transition of closed to open cells. Weather Research and Forecasting (WRF) model simulations from Zhou et al. (2018) indicate that moisture stratification and precipitation tend to increase horizontal cloud scales by enhancing updraft buoyancy via increased latent heating. Additionally, longwave radiative cooling near cloud top plays a crucial role in increasing horizontal cloud scales, and sub-cloud moist cold pools tend to respond to, rather than determine, mesoscale variability. A Lagrangian framework has been shown to be effective in capturing upstream conditioning on developing clouds (Lewis et al., 2023) as well as to be used to quantify cloud lifetime, and track changes in cloud microphysics associated with changes in aerosol concentration and meteorological conditions (Christensen et al., 2020, 2023).

The shortwave cloud radiative effect of transforming open to closed stratocumulus cells was estimated to be as large as 109 W/m$^2$ in a composite of 50 case studies from MODIS observations from Goren and Rosenfeld (2014). Goren and Rosenfeld (2014) decomposed the aerosol indirect effect into the Twomey effect (the enhancement in shortwave cloud albedo caused by increasing cloud droplet concentration for fixed changes in liquid water path), and rapid adjustments containing liquid water path and cloud fraction changes. These were estimated to be approximately 26%, 32%, and 42%, respectively. Here, we also quantify cloud water path and fraction adjustments but using a regression technique following Quaas et al. (2008) applied to kilometer-scale WRF model simulations of marine stratocumulus. We utilize a Lagrangian framework to capture the evolution of low-level clouds and examine how their cellular patterns change over time in order to answer the following research questions:

- To what extent does a change in aerosol concentration modify the area and spacing between cloud cells?

- How does the aerosol indirect radiative effect vary over diverse meteorological conditions?

- How does changing PBL and microphysics schemes affect the aerosol indirect effect?

- How do liquid water path and cloud fraction adjustments compare to the Twomey effect?

To answer these questions we first describe the details of the data sets used in this study (section 2), set up several case study experiments in WRF that utilize a Lagrangian framework (section 3), and conclude with an assessment of the aerosol radiative forcing (sections 4 and 5).

## 2  Observational Data

The U.S. Department of Energy Atmosphere Radiation Measurement (ARM) program has been providing continuous measurements of cloud properties at Graciosa Island in the Azores for over a decade. This location is ideal for studying mesoscale structure (Jensen et al., 2021), turbulence (Ghate and Cadeddu, 2019), and aerosol-cloud interactions (Zheng et al., 2022b; Christensen et al., 2023; Varble et al., 2023) in marine stratocumulus clouds. Ground-based measurements from ARM, aircraft measurements from the Aerosol and Cloud Experiments in the Eastern North Atlantic (ACE-ENA; Wang et al., 2022), and satellite observations from geostationary and polar orbits are used to evaluate WRF simulations of boundary layer clouds passing over Graciosa Island.

Significant progress in the process-scale understanding of aerosol-cloud interactions, facilitated by observational data from Graciosa Island, reveals that the seasonal cycle plays a significant role in aerosol activation. During winter, when the clouds are more decoupled and connected to stronger updrafts compared to summertime conditions (Wang et al., 2022; Zheng et al., 2022a), a higher fraction of accumulation mode particles tends to be activated. Despite higher activated aerosol fractions in winter, droplet number concentrations are lower due to less available aerosol compared to summer conditions (Wang et al., 2022). Large eddy simulations (LES) using the WRF model with spectral bin microphysics and dynamical downscaling from a 19 km horizontal resolution to a 300 m grid spacing (Wang et al., 2020) demonstrated that imposing aerosol plumes at observed aircraft heights significantly reduces the effective droplet radius ($ACI_r = \frac{\partial \ln(R_e)}{\partial \ln(N_{CCN})} \approx -0.11$) and increases the liquid water path ($ACI_l \approx +0.14$). These cloud microphysical changes may modify the dynamics in the planetary boundary layer differently between seasons. Consequently, our work focuses on characterizing the cloud fraction response from numerous summer and winter case studies provided by ACE-ENA, conducting an in depth investigation into mesoscale structural changes in clouds, and bridging the gap between cloud morphological changes and aerosol radiative forcing in low clouds.

### 2.1  Ground-based observations from ENA

Rain rate is retrieved using a laser optical OTT Particle Size and Velocity (PARSIVEL-2) disdrometer, which measures the instantaneous rainfall rate by quantifying the water flux from drops in 32 size bins (0 to 25 mm) and 32 fall velocity bins (0.2 to 20 m/s) falling to the surface. The retrieval has a 6% absolute bias with respect to reference gauges over a 1-min sampling interval (Tokay et al., 2014) as provided in the LDQUANTS value added product (Hardin et al., 2020).

Cloud top height and low-level cloud fraction are estimated from the active remote sensing of clouds (ARSCL) product (O'Connor et al., 2004; Kollias et al., 2016; Clothiaux et al., 2001), which combines vertically pointing Ka-band radar and lidar data to produce high-resolution time-height cross sections of cloud boundaries.

Bottom of atmosphere shortwave and longwave radiative fluxes are provided by the ARM best-estimate cloud radiation dataset (ARMBECLDRAD; Xie et al., 2010; Tang and Xie, 2020) in hourly intervals using measurements from a infrared radiation station. Temperature and specific humidity profiles containing 266 altitude levels are provided every minute by the Interpolated Sounding (INTERPSONDE; Troyan, 2013) product that combines observations from radiosondes, the microwave radiometer (MWR), and surface meteorological instruments.

The effective radius of cloud droplets and optical depth in single-layer overcast liquid-only clouds is determined using the multifilter rotating shadowband radiometer (MFRSR) at a wavelength of 415 nm (Turner et al., 2021). The retrieval process relies on the algorithm developed by Min and Harrison (1996) for atmospheric radiative transfer. If the MWR successfully retrieves liquid water path, then the effective radius is calculated based on the MWR and MFRSR data. However, if this information is not available we exclude it (occurring less than 30% of cases) from the analysis to avoid using fixed effective radius replacement values of 8 $\mu$m in the ARM product.

## 2.2 ACE ENA Flights

The ARM Aerial Facility Gulfstream-159 (G-1) research aircraft flew from Terceira Island in the Azores during two intensive operational periods (IOPs) that occurred from June to July, 2017 and January to February, 2018 during ACE-ENA. Deployments during both seasons are used to evaluate the vertical profile of the bulk liquid water content measured by the multi-element water content system (WCM-2000; Matthews and Mei, 2017). The multi-element water content measuring system utilizes a scoop-shaped sensor to measure total water content, capturing both liquid and ice phase hydrometeors. It incorporates two heated wire elements (021-wire and 083-wire), exposed directly to the airstream, along with a reference element exposed to the airflow but not to condensed water. Following the approach of Miller et al. (2022), we adopt the WCM-2000 system due to its favorable agreement in liquid water content measurements compared to the Fast Cloud Droplet Probe and Two-Dimensional Stereo particle imaging probe measurement systems. The condensation particle counter (CPC) measures the number concentration of aerosols from 10 nm to 3 microns under-kinetic mode. Aerosol concentration uncertainties are approximately 15% (Fan and Pekour, 2018). The cloud condensation nuclei concentration is obtained from the CCN-200 particle counter aboard the G-1 aircraft providing CCN at approximately 0.2% supersaturation every second (i.e., N_CCN_1 as discussed in Uin and Mei, 2019). To compare aerosol properties in clear-sky conditions with the WRF model, we select aircraft samples within a $1° \times 1°$ region centered around the ARM site at 13:00 UTC $\pm$ 1.5 hr and below 2 km altitude, excluding those with cloud water content.

## 2.3 Satellite observations

Cloud-top effective droplet radius ($R_e$) and cloud optical thickness ($\tau_c$) are retrieved from the 1.6, 2.1, and 3.7-$\mu$m channels; cloud top temperature, pressure, and height are retrieved from longer-wavelength thermal channels on the Moderate Resolution Imaging Spectroradiometer (MODIS) instrument. These data, retrieved from the collection 6.1 cloud product (Platnick et al., 2017), are at a 1-km pixel-scale resolution at nadir from satellites Terra and Aqua, passing over the region at approximately 10:30 am and 1:30 pm local time, respectively. Of the three spectral channels used for $R_e$ retrievals, the sensitivity of the

3.7-$\mu$m channel is weighted closest to the cloud top, primarily due to the relatively strong absorption of water vapor at this wavelength (Platnick, 2000). Because errors in the adiabatic droplet number concentrations using the 3.7-$\mu$m channel are considerably smaller than in the other bands (Grosvenor et al., 2018), we choose to use it for this study.

Imagery from the Geostationary Operational Environmental Satellite (GOES) Advanced Baseline Imager (ABI) of the National Oceanic and Atmospheric Administration (NOAA) GOES-R series satellite (Pinker et al., 2022) is utilized to aid in visualizing the evolving characteristics of mesoscale cloud structures along Lagrangian trajectories. Full-disk images covering the entire region are made available every 15 minutes. These images have spatial resolutions of 0.5 km at nadir for the 0.64-$\mu$m visible channel and 2 km for the 3.9-$\mu$m and 11-$\mu$m channels.

The Clouds and the Earth's Radiant Energy System (CERES) Synoptic (SYN1deg-1Hour) edition 4.1 product (Rutan et al., 2015) provide similar cloud top retrievals to MODIS using similar algorithms (e.g. the MODIS collection 5 product) as well as top and bottom of atmosphere shortwave and longwave radiative fluxes that are gridded globally at $1 \times 1°$ every hour through combining multi-spectral retrievals from a network of 16 geostationary satellites as well as the CERES instruments on Terra, Aqua, and Suomi National Polar-orbiting Partnership.

## 3   Methodology

Figure 1 depicts a 4-step procedure used to initialize and run the Weather Research and Forecasting (WRF) version 4.3 model (Skamarock et al., 2021) in a Lagrangian framework. This technique uses an inner nest that moves through the WRF model (outer domain) at specified time-steps. First, the Hybrid Single-Particle Lagrangian Integrated Trajectory (HYSPLIT; Stein et al., 2015) version 5 model is used to calculate a 6-hour back and a 6-hour forward trajectory using the Modern-Era Retrospective analysis for Research and Applications, version 2 (MERRA-2; Gelaro et al., 2017) reanalysis meteorological data. Trajectories are calculated from the middle of the planetary boundary layer (determined in HYSPLIT ). This height has been shown to be representative for tracking the general flow of boundary layer clouds over the ocean (Christensen et al., 2020; Kazil et al., 2021; Christensen et al., 2023). Back trajectories are initialized at the Graciosa Island ARM site at 10 am local time before the Terra (morning at 10:30 am) and Aqua (afternoon at 1:30 pm) MODIS overpass times. Forward and backward trajectories are initialized at Graciosa Island and run for 6-hours. These trajectories are combined to form a 12-hour trajectory starting from the tail of the back trajectory and ending at the tail of the forward trajectory. This method ensures that the airmass transits over the ARM site.

### 3.1   WRF Modeling

Nested simulations are performed using the WRF model (Figure 1 box 2). The outer (static) domain is $12° \times 12°$ and is centered over the ARM site on Graciosa Island. The region is large enough to span the entire length of the back and forward Lagrangian trajectories. The outer domain has a horizontal grid spacing of 4 km and a vertical grid that is log-stretched where the spacing is approximately 50 m near the surface and increases to 150 m throughout the PBL to the top of the model at 20 km. The model

time-step is 10 s. The outer domain is used to characterize the large-scale meteorological flow and boundary conditions for the inner domain.

The inner domain allows for convection-permitting scales and moves along the HYSPLIT trajectory using the multi-incremental 4D-Var system which allows for translating (moving) nests within WRF (similar to vortex tracking for hurricanes as described in Zhang et al., 2014). WRF was compiled using preset moves to permit higher spatial resolution simulations within the inner domain which is is computationally more efficient than high resolution across the entire outer domain. The inner domain translates in time (across the outer domain) according to the pre-computed locations using HYSPLIT. Given the spatial scales of typical cellular maritime cloud organization (30 to 40 km; Wood, 2005), the inner domain is spatially large enough to capture the largest scales of variability spanning approximately $200 \times 300$ km$^2$ with a horizontal grid-spacing that is 5 times finer than the outer nest (800 m) with the same vertical resolution.

Boundary conditions are initialized and updated every 6 hours during simulation using reanalysis data from MERRA-2 which is spatially gridded at 0.5-degree resolution with 72 vertical levels and provided every 6 hours. We have tested WRF using other meteorological data sets (see Text S1 and Fig. S1 for details) and find that the choice of the reanalysis product does not significantly alter the results. To coincide with earlier work (Christensen et al., 2023) we use MERRA-2 to drive the WRF boundary conditions for this study.

We use a 6-hour spin-up period to allow sufficient time for the cloud properties to reach steady state. After this period, the inner two-way nest begins to move within the WRF model according to the HYSPLIT trajectory computed using the same reanalysis product as that was used to drive the WRF model. The simulations are performed with the aerosol-aware Thompson bulk microphysical parameterization scheme (Thompson and Eidhammer, 2014) with explicit cloud droplet nucleation treatment following Köhler activation theory. Look up tables generated from parcel modeling are used to provide the cloud droplet number concentration based on predicted temperature, vertical velocity, number of hygroscopic aerosol particles also referred to as 'number of water friendly aerosols' (NWFA), and predetermined values of hygroscopicity parameter and aerosol mean radius. Aerosol sensitivity experiments follow the same approach as described in Thompson and Eidhammer (2014) in which the input mass mixing ratio of each aerosol species (dust, sea salt, black and organic carbon, and sulfate aerosols) is obtained from GOCART and is converted to NWFA concentration using assumed lognormal distributions with characteristic diameters and geometric standard deviations taken from Chin et al. (2002) (their Table 2). Next, we modify the NWFA concentration profile climatology averaged over 7 years using the following scale factors: 0.01 (pristine), 0.1 (clean), 1.0 (control), and 10.0 (polluted) for each experiment, respectively (Figure 1 box 3). Note, that no changes are made to the assumed aerosol chemical species composition, hygroscopicity parameter (0.4 in experiments performed in this research), and aerosol mean radius (0.04 $\mu$m). These scale factors significantly affect the NWFA concentration as shown in Figure S2. Lower condensation particle concentrations (CPC) in cloud-free air sampled by the aircraft suggest that the control simulation of NWFA may be more polluted than the observations on this particular day and across seasons (Figure S3). However, the CPC and NWFA serve as a rough comparison as the characteristics (namely the size distribution and hygroscopicity) of these two quantities may differ. As discussed later, cloud droplet number concentrations are also affected by NWFA with median values broadly approaching 20, 50, 250, and 450 cm$^{-3}$ for our pristine (N1), clean (N2), control (N3), and polluted (N4) aerosol experiments, respectively.

The Level-3 Mellor-Yamada-Nakanishi-Niino (MYNN3) PBL scheme (Nakanishi and Niino, 2009) predicts TKE and other second-order moments within the PBL. The Rapid Radiative Transfer Model for GCMs (RRTMG) specifies the size of hydrometers and utilizes the correlated-k approach to calculate fluxes and heating rates accurately (Iacono et al., 2008) and efficiently through its use of a Monte-Carlo Independent Column Approximation technique (Pincus et al., 2003). The simulations utilize the Noah land surface model (Barlage et al., 2010) as well as the Tiedtke cumulus scheme (Zhang et al., 2011).

Model evaluation (Figure 1 box 4) is carried out using output from the WRF-Solar model (Jimenez et al., 2016), which passes the effective radius of cloud particles from the microphysics to the radiation parameterization scheme (Thompson and Eidhammer, 2014), impacting cloud albedo and enabling quantification of the aerosol indirect effect (Thompson et al., 2016). WRF-Solar includes a solar diagnostics package that outputs several two-dimensional variables, including cloud fraction, liquid effective droplet radius, optical thickness, and liquid water path. The liquid water path is computed from the effective radius and optical thickness quantities, i.e., $LWP = \frac{2}{3}\tau_c R_e$ where $\tau_c$ is the cloud optical thickness, and $R_e$ is the effective droplet radius (Stephens, 1978). These quantities (that are weighted towards the cloud top) have been shown to be comparable with MODIS observations (Otkin and Greenwald, 2008). A summary of the model setup is listed in Table 1.

## 3.2 Case Studies

Figure 2 shows our selected case studies. Days are selected based on the following criteria: 1) a dearth of high-level cloud over the trajectory for optimal comparison with satellite retrievals, 2) aircraft measurements coinciding with intensive operation periods (IOP) 1 (6/25/2017 - 7/25/2017) and 2 (2/1/2018 - 2/25/2018), and 3) diverse meteorological conditions to study the impacts of precipitation, atmospheric stability, and free tropospheric humidity states on aerosol-cloud interactions. Across the experiments, the height of the PBL top varied from 600 m to 1710 m and the surface air temperature varied from $13 - 22°C$ as determined by meteorological soundings averaged over the entire day. Daily total accumulated precipitation from the disdrometer varied from 0 - 4 mm. A wide range of cloud patterns were observed including disorganized (small, isolated clouds or clouds with no discernible pattern), homogeneous (solid cloud deck with no discernible pattern), closed-cells (cells filled with cloud), open-cells (cells where the center is devoid of cloud). These classifications are broadly inferred using the definitions described in Wood and Hartmann (2006). Table 2 lists key quantities of interest for the cases displayed in Figure 2. It is noteworthy to mention that while we aim to select cases which did not have ice cloud in the observations, the WRF model sometimes simulated them above the boundary layer (7/18/17) and within the boundary layer during two of the wintertime IOP case studies (1/24/18 and 1/25/18). Potential impacts of simulated ice cloud on the analysis are discussed in subsequent sections.

## 3.3 Lagrangian Framework and Dataset Integration

Figure 3 shows the evolution of shallow clouds in the Lagrangian trajectory for the lightly drizzling day of 7/18/2017. This case study forms the backbone for many of the inter-comparisons made throughout this work due to the distinct closed cell features and persistence of the stratocumulus cloud deck throughout the day. Satellite retrievals from GOES and MODIS are aggregated over a $1 \times 1°$ region (yellow box) during each time-interval (15 minutes) along the trajectory. CERES gridded-data

is interpolated in space and time to the same trajectory grid-box. WRF simulations at roughly km-scale are aggregated over the same region and timescale as the Lagrangian trajectory. Both the observations and simulations show persistent closed cell clouds throughout the day. These clouds produce very light drizzle as indicated by the Ka-band radar (Fig. 2e) and disdrom-
225 eter measurements at Graciosa Island (Table 2). An evident wake island effect is observed and simulated in the downstream region from the Azores. In general, the low-level flow and horizontal displacements of the clouds are well captured using the Lagrangian framework as depicted in Movie S1.

## 4 Results

In the first part of the analysis we quantify the effect of aerosol changes on the mesoscale structure of clouds (i.e. size and
230 distance between cloud cells) and associated radiative impacts from an ensemble of 40 WRF simulations spanning 10 different case studies with 4 varying aerosol concentrations (a set of 4, for each case study day) offering diverse meteorology and cloud types. This particular set of simulations uses MYNN3 and Thompson (aerosol-aware) PBL and microphysics schemes, respectively. In the second part (section 4.3.2), we assess and quantify variations in the aerosol indirect effect on case study day 7/18 by employing different PBL and microphysical scheme choices across 26 WRF experiments.

### 4.1 Impact of aerosol on the mesoscale structure of clouds

Cloud objects are detected using a watershed technique, following the methodology described in Wu and Ovchinnikov (2022). Because the standard WRF model output does not include simulated channel reflectances for MODIS, comparisons are made based on the $LWP$. The only difference between Wu and Ovchinnikov (2022) and our study is that we use $LWP$ instead of the MODIS reflectance. As $LWP$ scales well with the visible cloud albedo (Stephens, 1978), the replacement of $LWP$
for visible reflectance is suitable after thresholds have been linearly scaled. Moments of the $LWP$ distributions have been used for cloud classification of marine stratocumulus in several studies (e.g., see Wood and Hartmann, 2006; Zheng et al., 2018). The segmentation procedure initially smooths the $LWP$ field to remove random field variations while preserving object boundaries using a two-dimensional Gaussian filter with a kernel standard deviation of 250 g m$^{-2}$. Next, cloud objects are detected using a watershed technique. A centroid is assigned to each cloud object based on the distribution of cloudy pixels
with $LWP$ greater than 100 g m$^{-2}$. Cloud objects are formed if a common interface is shared. An edge weight is computed, and if the area-weighted mean difference between pixels along the interface is smaller than 4 g m$^{-2}$ the two objects are merged and a new centroid is assigned to the object.

To determine the spacing between cloudy object centers, we compute the distance of each cloud object centroid to all other centroids and select the minimum distance (i.e., $D_c$). Due to variable sizes of the cloud objects, we also compute the distance
of all edge pixels of an object to all of the edge pixels of all other objects and select the minimum distance ($D_e$). This latter method provides an estimate of the closest distance between neighboring cloud object boundaries, thus removing the effect of cloud fraction on distances between clouds that is not accounted for with cloud object centroids.

Cloud objects are identified in WRF (Fig. 4b) every 15 mins along the trajectory and in MODIS at the Terra and Aqua overpass times (Fig. 4d). Cloud area ranges from about 1 - 500 km$^2$ in WRF and MODIS (Fig. S4a). The majority are at scales less than about 10 km, a result similarly found in Wu and Ovchinnikov (2022) and Wood and Hartmann (2006), based on power spectral analysis of the spatial variance in $LWP$. The distance between cloud object centroids is similar between MODIS and WRF with a mean value of approximately 12.1 km and median value of roughly 10.7 km for this particular case study (Fig. S4b).

The size and spacing between cloud objects is to some extent dictated by the background aerosol concentration. Figure 5a and b show that the average cell area and spacing between object centroids increases as the background aerosol concentration increases. The distance between cloud edges decreases as the aerosol concentration increases (Figure 5c). This is evident when comparing 'snapshots' of the pristine and polluted experiments taken at the same time (Fig. 5d-e). The cloud objects are spreading away from each other but they are also becoming larger and filling the gaps between clouds as aerosol loading increases. Similar behavior is found on 7/15/2017 (as depicted in Fig. S5) and generally across all case studies (discussed in section 4.3).

To characterize uncertainty and determine whether this relationship is robust, a sensitivity test of the segmentation algorithm is performed over a range of minimum $LWP$ thresholds for defining cloud object edges spanning 1 to 500 g m$^{-2}$. Figure S6 shows that the area of the cloud objects become larger with increasing aerosol concentration. This response is robust across the full range of $LWP$ threshold values. The largest sensitivity of this relationship occurs around 200 g m$^{-2}$. This unique threshold $LWP$ value is also a turning point for which further increases in $LWP$ decrease the number of detected cloud objects, which impacts cell separation distance. Furthermore, the cloud fraction is larger under polluted conditions and this relationship is robust for each minimum LWP threshold value (Fig. S6d). As 100 g m$^{-2}$ forms roughly the midpoint value we select this representative threshold for segmenting clouds in this analysis.

## 4.2 Aerosol-cloud interactions

Two case studies, one with lightly precipitating clouds and another with heavier precipitating clouds are examined in detail during the summertime IOP period for quantifying the effects of aerosol on precipitating and lightly-precipitating clouds.

### 4.2.1 Lightly-Precipitating clouds

On 7/18/17 closed-cell type clouds were found in the vicinity of the Azores. The clouds produced a light amount of precipitation where only approximately 0.02 mm was recorded in the distrometer measurements from ARM. Aircraft measurements of the cloud water content on this day fit within the range of variability simulated for clouds in the WRF model (Figure 6a). Cloud tops from the aircraft measurements imply that WRF simulates a slightly deeper than observed boundary layer by approximately 200 m. We find reasonable agreement between MODIS, CERES, and ARM data sets with the WRF simulations (Fig. 7). Cloud optical depth and radiative fluxes tend to agree more closely with the clean and control WRF experiments. The agreement not being closest with the control experiment may be indicative of the following issues: 1) a bias in the climatological aerosol concentrations (being too high), 2) the Thompson scheme may be nucleating too many aerosols, or 3) scavenging rates are

not large enough. Despite these differences, the chosen schemes resolve essential characteristics of a realistic boundary layer based on the reasonable agreement in the cloud relevant properties.

Rain water mixing ratio, also forming closer to cloud top in the cleaner experiments, decreases up to an order of magnitude as background aerosol concentration increases (Figure 6b). A modest increase in cloud water content and cloud water mixing ratio is found in the more polluted simulations throughout all levels in the cloud. This result is consistent with the indirect effect using the Thompson microphysics scheme described in Thompson and Eidhammer (2014). An increase in aerosol concentration also results in smaller cloud droplet effective radius (Fig. 7a), larger cloud optical thickness, larger liquid water path, and larger droplet concentration (Fig. 7b,c,d); cloud-top quantities are obtained from WRF-Solar. The more polluted aerosol experiments with optically thicker clouds result in more reflected solar radiation at the top of the atmosphere and less incoming solar radiation at the bottom of the atmosphere despite having slightly lower cloud tops. The slightly elevated cloud tops in the more pristine simulation also have elevated cloud bases and are more decoupled from surface moisture. Nonetheless, all simulated cloud top heights are within the range of variability in the ARM and satellite observations.

Cloud properties tend to vary over the course of the trajectory with increasing cloud optical thickness, liquid water path, and cloud top height. This is accompanied by an increase in sea surface temperature and more unstable boundary layer conditions along with rising lifted condensation level and decreasing free tropospheric humidity (Fig. S7). A deepening boundary layer is expected given the warming sea surface temperature (Eastman et al., 2016) but despite the changing meteorological conditions over the trajectories, the cloud alterations attributed to changes in aerosol loading remain systematic throughout the 12-hour period.

### 4.2.2 Precipitating clouds

In comparison to the previous case study, the boundary layer on 7/15 is about 750 m deeper and the accumulated rainfall is significantly larger; 3.9 mm. Much like the previous light drizzle case study, the properties of precipitating clouds on 7/15/17 also broadly fit within the range of variability in cloud water content as measured by aircraft observations (Figure 8) and LWP by satellite and ARM retrievals (Fig. S8c). Simulated cloud top height tends to be higher than the observations during the afternoon hours. Figure 2d reveals more vigorous clouds, as indicated by relatively large radar reflectivities during early morning and late afternoon periods outside the trajectory timeframe. This difference could contribute to the observed mismatch between simulated and actual cloud top heights. Despite this, in the control simulation, peak cloud water contents are approximately 40% larger, and peak rainwater mixing ratios are about 90% larger on 7/15 (precipitating case study) compared to 7/18 (drizzling case study). Furthermore, the cloud water content increase due to increasing aerosol concentration is significantly larger on 7/15 compared to 7/18.

Simulations with elevated concentrations of aerosols have larger cloud top shortwave and longwave radiative cooling rates. The net radiative cooling rate decreases from approximately -10 K/d in the clean simulations to -30 K/d in the more polluted simulations (Figure 9). Mean vertical and horizontal wind velocity near the cloud top also tends to be larger in the more polluted simulations. Vertical velocity variance and turbulence throughout the boundary layer tend to be larger in the more polluted simulations. Vertical profile shapes of these quantities are similar, albeit less pronounced, on 7/18 (Figure S9). Stronger

updrafts in the more polluted simulations where radiative cooling rates are larger coincide with larger lateral displacements near the base of the inversion and may be partially responsible for causing the significant widening of the clouds and increased cloud fraction.

To account for the turbulence of the convective eddies at 800-m grid spacing (in the so-called "gray-zone" where eddies in the PBL are partially resolved; Shin and Dudhia, 2016), TKE is also provided using the 3D resolved winds (Figure 9i) following the equation: $TKE = \frac{1}{2}(u'^2 + v'^2 + w'^2)$, where $u'^2$, $v'^2$, and $w'^2$ are the variances of the winds computed over $3.2 \times 3.2$ km$^2$ regions. In our simulations, the resolved TKE is not very sensitive to changes in the averaging scale in which the 3.2, 6.4, and 12.8 km scales show similar magnitude within the cloud layer. While the TKE computed using the resolved winds shows a relative increase near cloud top hinting at a better connection to the cloud top radiative flux profile compared to the subgrid TKE output from MYNN3, this is a relatively weak relationship compared to large eddy simulations of stratocumulus (e.g., see McMichael et al., 2019), which may suggest further refinement is needed in connecting these processes within the MYNN Eddy-Diffusivity Mass Flux (EDMF) scheme (Olson et al., 2019). Possible implications of the relatively weak mixing on the liquid water path and cloud fraction adjustments are discussed in further detail in the conclusions section.

Additional tests are carried out at 1 km horizontal grid spacing to determine the relative roles of cooling caused by rain drop evaporation (by setting the temperature and moisture tendencies caused by changes in rain mass evaporation in the Thompson microphysics scheme to zero), cloud radiative effect (setting icloud=0 in the namelist file), and the cumulus scheme (by turning it off) on the results. Rain evaporation below cloud base stabilizes the atmosphere, producing decoupling and less turbulence (Wood, 2012). However, Figure S10 shows that turning off rain droplet evaporation results in only a small relative change in cloud and rain mixing ratios, radiative cooling, and turbulence. Turning off the radiation to the clouds significantly decreases turbulent mixing, cloud top height, and rain water mixing ratio. Similarly, turning off the cumulus scheme significantly decreases cloud and rain water mixing ratio and radiative cooling rates.

Figure S11 illustrates the impact of sensitivity experiments on aerosol effects on cloud properties. In general, an increase in aerosol concentration enhances cloud fraction, liquid water path, and cloud area extent as aerosol loading increases. Turning off cloud interactions with radiation removes the effects of changes in cloud radiative heating and cooling, but clouds still expand (albeit less so) simply due to precipitation suppression by aerosols. This may indicate that low-clouds expand due to precipitation suppression, through reducing the magnitude of the primary cloud sink; next changes in radiative effects cause further increases in cloud fraction (approximately 100% more based on Fig. S11). This cloud radiative feedback suggests a notable contribution to promoting the initial cloud expansion via precipitation suppression by aerosol. For removal of rain evaporation, the precipitation effect on PBL turbulence is turned off. While this no longer conserves energy (which is unavoidable in such sensitivity tests) we continue to simulate strong cloud expansion due to increased aerosol concentration and this is largely due to the suppression of precipitation and growth of the cloud.

### 4.3  Aerosol-cloud interactions across 10 case studies

A suite of aerosol experiments spanning 10 case studies with varying meteorological conditions provides 40 WRF simulation experiments to examine aerosol indirect radiative effect across a range of meteorological and cloud conditions. These case studies are summarized in Table 2.

#### 4.3.1  Aerosol indirect radiative effect

The aerosol indirect radiative effect is calculated from the change in the top of atmosphere outgoing shortwave radiative flux caused by a change in $N_d$ and can be written as

$$RE_{aci} = -\overline{F^{\downarrow}}\phi_{atm}\overline{\frac{f_c\alpha_c(1-\alpha_c)}{3N_d}}\left(1 + \frac{5}{2}\frac{\Delta\ln L}{\Delta\ln N_d} + \overline{\frac{3(\alpha_c-\alpha_{sfc})}{\alpha_c(1-\alpha_c)}}\frac{\Delta\ln f_c}{\Delta\ln N_d}\right)\overline{\Delta N_d} \tag{1}$$

where, $F^{\downarrow}$ is the top of atmosphere (TOA) incoming solar radiation, $\phi_{atm}$ is the transfer function that accounts for the transmissivity (reflection and absorption) of the non-cloudy air above the surface and takes an average value of 0.7 (Diamond et al., 2020), $f_c$ is the cloud cover fraction, $\alpha_c$ is the cloud albedo, $N_d$ is the droplet concentration, $L$ is the liquid water path, and $\alpha_{sfc}$ is the surface albedo. The full derivation, based on Quaas et al. (2008) and Christensen et al. (2023), is described in Text S1.

Quantities in equation 1 are obtained in hourly intervals over a $1° \times 1°$ domain moving along the trajectory. The $\Delta$ symbols denote differences between aerosol experiments of varying aerosol concentrations. There are six possible pairs which include, polluted $-$ control $\Delta$(N4−N3), polluted $-$ clean $\Delta$(N4−N3), polluted $-$ pristine $\Delta$(N4−N1), control $-$ clean $\Delta$(N3−N2), control $-$ pristine $\Delta$(N3−N1), and clean $-$ pristine $\Delta$(N2−N1). $\overline{F^{\downarrow}}$ is the daily-mean solar insolation, $\overline{\frac{f_c\alpha_c(1-\alpha_c)}{3N_d}}$ and $\overline{\frac{3(\alpha_c-\alpha_{sfc})}{\alpha_c(1-\alpha_c)}}$ are computed from mean quantities of the paired aerosol experiments, and $\overline{\Delta N_d}$ represents the mean difference in cloud droplet number concentration between paired aerosol experiments. By using a wide range of aerosol concentrations we aim to capture variability in ACI but acknowledge that non-linearity in the relationship between cloud variables with $N_d$ may be missed from the use of only 4 aerosol experiments.

Figure 10 shows the relationship of key variables as they change in response to increasing background aerosol concentrations in the WRF model. In most cases, there is good agreement in the sign of the response across diverse case studies. An increase in aerosol concentration enhances the top of atmosphere reflected sunlight, cloud fraction, liquid water path, cloud optical thickness, and cloud object area. A robust decrease in droplet effective radius is also evident. While responses are mostly consistent, the magnitude can vary substantially. Cases where significant precipitation occur (7/15 and 1/25) exhibit the largest increases in liquid water path, cloud optical thickness, and cloud object area. Days having light rain (7/18, 7/6, 7/12) or no measurable rain (6/30, 1/24) have significantly weaker responses by comparison. Figure 11a,b shows the effect of precipitation on the liquid water path and cloud fraction aerosol adjustments. While there is some scatter across experiments, this result generally agrees with Chen et al. (2014); an increase in aerosol concentration has a stronger radiative effect on precipitating clouds compared to non-precipitating clouds due to the suppression of precipitation causing cloud water to increase. While

drizzle suppression reduces scavenging of cloud droplets and goes into spreading the cloud vertically, the horizontal spreading of the clouds through increased cloud object area is highly significant.

The Twomey radiative effect estimated as -13.7 $\pm$ 9.3 W m$^{-2}$ with a range extending from -4.1 W m$^{-2}$ to -29.9 W m$^{-2}$
across case studies. This estimate is based on the daily-mean solar insolation, which at this location, can vary significantly between winter and summer IOP periods. Note, this estimate is the radiative effect, not the radiative forcing, and hence does not include the changes in aerosol concentration attributed to anthropogenic sources (i.e. the present-day minus pre-industrial values). The radiative effect is estimated from 6 different aerosol experiment pairs (discussed above) that have a wide range of aerosol concentrations (as shown in Fig. S2 and Fig. S3). The cloud properties and radiative effects associated with each case
study are listed in Tables S1 – S10. The quantification of the modeled sensitivity in the cloud radiative effect to changes in cloud droplet concentration are similar to those found in satellite observations of ship tracks (Christensen and Stephens, 2012; Goren and Rosenfeld, 2014).

To make the results more intuitive, Table 3 lists the ratios of the liquid water path and cloud fraction adjustments scaled by the Twomey effect. These enhancements range from 10 - 150 % for the $LWP_{adj}$; a result that is similar to that found across
multiple GCM experiments in Gryspeerdt et al. (2020) and in the observations of Goren and Rosenfeld (2014). Consequently, our findings approach the upper limits of these adjustments possibly due to a weak connection between entrainment mixing and cloud top radiation from the use of km-scale models (discussed further in the conclusions). During both IOP periods we find that the largest indirect radiative effects tend to coincide with the largest daily precipitation rates (Figure 11c). These cases are also consistent with those which show the largest cell area growth as a function of aerosol loading (Figure 10).
The cloud object area expansion relationship is not as strong during the wintertime IOP period. Figure S12 reveals the presence of ice on 1/24/18 and 1/25/18, and intriguingly, the Twomey effect and rapid adjustments exhibit comparable agreement in these cases, as seen in the warm cloud case study days (Figure 10). Although the Thompson microphysics scheme considers ice multiplication from rime-splinters through the Hallett–Mossop process (Hallett and Mossop, 1974), a phenomenon known to lead to cloud morphology breakup and alteration, accompanied by enhanced precipitation (Abel et al., 2017; Eirund et al.,
2019), we haven't altered ice-friendly nuclei concentrations in this study. Modifying such concentrations could offer additional insights into aerosol-ice cloud interactions in future research.

### 4.3.2 Impact of changing PBL and microphysics schemes

We devise a set of sensitivity experiments where the microphysics and PBL schemes are varied to assess the uncertainty of modeling boundary layer clouds and ACI. These simulations use the double-moment Morrison microphysics (Morrison
et al., 2005) scheme with fixed cloud droplet number concentration. For the Morrison scheme, we used fixed droplet number concentrations with values of 20, 80, 320, and 1020 cm$^{-3}$ for our pristine (N1), clean (N2), control (N3), and polluted (N4) aerosol experiments, respectively. Values for the more polluted runs were increased to coincide with the scale factors used in the Thompson (aerosol-aware) scheme for simulating similar values of the cloud droplet number concentrations. The additional PBL schemes for testing use the non-local Yonsei University (YSU; Hong et al., 2006) or local Mellor–Yamada–Janjic (MYJ;
Mellor and Yamada, 1982) closure flux models. These schemes have differences in vertical mixing strength which affect

entrainment of dry air from above the PBL and can impact cloud properties differently depending on the scheme chosen (Hu et al., 2010). A summary of each sensitivity experiment is listed in Table 4. Note, running the Morrison microphysics scheme with fixed droplet number concentration does not allow for a full positive aerosol-cloud-precipitation feedback cycle as simulated in some LES simulations (e.g., Yamaguchi et al., 2017). This has been shown to have a significant influence on the mesoscale structure of clouds, and hence, cloud fraction (Goren et al., 2019; Diamond et al., 2022), potentially having a significant impact on the net radiative effect in this sensitivity study.

Figure 12 shows a comparison of WRF simulated $LWP$ with satellite observations from several different microphysics and PBL schemes for the 7/18/2017 case study. All schemes simulate boundary layer cloud in a similar geographic region as that observed by satellites. However, some of the WRF schemes under-predict $LWP$ and cloud top heights, in particular the YSU and MYJ PBL schemes (as shown in Fig. S13) with respect to MODIS. The MYJ scheme tends to produce smaller cloud cells containing much smaller liquid water paths compared to the YSU and MYNN3 schemes. In general, most of the simulations reproduce the vertical profiles of temperature, humidity, wind speed, and wind direction compared to ARM radiosonde measurements (Fig. S14) but with a slightly elevated capping inversion and dew point temperature. Overall, we find the best agreement with the Thompson and MYNN3 PBL schemes regarding how close the cloud and atmospheric state compare to the observations.

To test the impact of using different schemes in WRF on the aerosol indirect effect, four aerosol experiments are carried out for each model configuration, yielding a total of 24 simulations to quantify the range of variability in aerosol indirect effect for the case study occurring on 7/18/17. Due to computational constraints, we ran these simulations at a lower spatial resolution (3 km grid spacing for the inner nest) using 99 vertical levels. Here we exclude cloud area changes in the analysis due to the poorer ability of the model to simulate these structures at lower resolution and focus more on the microphysical changes across these model configurations instead.

Figure S15 shows the aerosol perturbations of various cloud properties for each of the six WRF configurations. Like before, all simulations show that an increase in aerosol concentration results in an increase in the reflected solar radiation, a reduction in cloud droplet effective radius, and an increase in cloud optical depth. The lower simulation resolutions produce similar sensitivities compared to the higher resolution simulation runs. For example, $\Delta \ln \tau_c / \Delta \ln N_d$ for the higher resolution run is $0.55 \pm 0.12$ and $0.48 \pm 0.15$ for the lower resolution run. It is noteworthy that the liquid water path and cloud thickness responses are negative in some of the configurations; however, a t-test indicates that there is not a significant difference from zero. The variations ($\sigma RE_{aci} = \frac{\sigma_{err}}{RE_{aci}}$, estimated from the standard error, $\sigma_{err}$, computed from the standard deviation normalized by the square root of the 10 cases divided by the total indirect effect effect) across experiments is approximately $\pm$ 30%. Given this range of variability in the indirect effect, we infer the microphysical cloud responses are robust across a wide range of possible model configurations. Thus, variations larger than this level in analyses of the 10 case studies with the Thompson and MYNN schemes are likely to be more related to meteorological and cloud state modulations as opposed to these particular chosen WRF schemes.

## 5 Conclusions

We devised a series of realistic WRF simulations using boundary conditions from MERRA-2 reanalysis to simulate PBL clouds as they pass over Graciosa Island in the Azores during the ACE-ENA field campaign. Kilometer-scale simulations were carried out within an inner-nest that moves along the Lagrangian flow of the PBL, making higher resolution simulations computationally feasible for studying aerosol-cloud interactions. The Lagrangian framework allows for the analysis of an evolving cloud field over time, although, for relatively short-timescales like those used here the aerosol responses were roughly

consistent along the length of trajectories. Cloud water content, temperature, humidity, and wind profiles were in the range of acceptable uncertainty as determined by comparison with aircraft observations and radiosonde measurements from Graciosa Island. WRF-simulated cloud microphysical properties and radiative fluxes were generally in closer agreement to the clean (not control) experiments. This result suggests that the baseline NWFA concentration are biased high at Graciosa Island. With the exception of WRF simulating higher cloud tops during the afternoon compared to MODIS and ARM, the simulated cloud

and radiative properties in general tend to fit within the range of observed uncertainty.

With these simulations, we addressed the following research questions:

– **To what extent does a change in aerosol concentration modify the area and spacing between cloud cells?** An increase in aerosol concentration suppresses precipitation, causing liquid water content and liquid water path to increase throughout the PBL. Through applying the cloud segmentation watershed algorithm developed by Wu and Ovchinnikov

(2022) we find that cloud water mass is re-distributed through the PBL horizontally and in some cases vertically through the expansion of the clouds. This is accompanied by a decrease in clear skies between clouds. The suppression of drizzle through an increase in aerosol concentration results in more cloud water. The cloud top radiative cooling rate, turbulent kinetic energy, and vertical velocity variance increased in strength under polluted conditions. Larger horizontal winds near the cloud tops was typically found in the simulations with more aerosol. Through this process, the additional water

(not lost through drizzle) in polluted clouds is re-distributed both vertically as well as horizontally. This results in the expansion of cloud cells.

– **How does the aerosol indirect radiative effect vary over diverse meteorological conditions?** The sign of the aerosol indirect radiative effect is robust across all case study days. They all exhibit liquid water path and cloud fraction increases with increasing aerosol concentration, a similar result also found in the WRF simulations of Zheng et al. (2022a). As

found in previous studies (e.g., Chen et al., 2014), the strength of the radiative effect is strongly tied to the occurrence of precipitation. We find that the cloud area expansion is greater in environments that support deeper boundary layers with heavier precipitation and the magnitude is generally smaller in case studies with less background precipitation.

– **How does changing PBL and microphysics schemes affect the aerosol indirect effect?** A set of six WRF configurations using three different PBL and two different microphysics schemes revealed robust cloud responses to changes in

aerosol concentration. The range of variability on total indirect effect across configurations was approximately 30%. We conclude that the choice of valid WRF schemes plays less of a role on the indirect effect (at least from these configu-

rations for one case study) than the impact of precipitation on aerosol-cloud interactions where the variations are larger across the 10 case studies.

– **How do liquid water path and cloud fraction adjustments compare to the Twomey effect?** Aerosol radiative effects were decomposed into contributions from the Twomey effect and liquid water path and cloud fraction adjustments. The liquid water path and cloud fraction adjustments scale as 74% and 51% increases relative to the Twomey effect, respectively. These adjustments are largest where an increase in aerosol can have a larger impact on drizzle suppression and cloud water path enhancement. Our simulation estimates of the scaled adjustments are larger but within the range of uncertainty estimated from satellite observations (Goren and Rosenfeld, 2014).

As computation power increases, km-scale models employed with PBL schemes (similar to ours) will increasingly be used to quantify aerosol-cloud interactions at global-scales with increasing complexity (Terai et al., 2020). Kilometer scale models have been shown to successfully simulate the properties and mesoscale structure of stratocumulus. Chen et al. (2022) used WRF with 1km grid spacing to simulate the roll structure and transition of stratocumulus and cloud streets by gradients in sea surface temperature. Saffin et al. (2023) utilized the Met Office Unified Model to simulate cloud transitions observed during the ATOMIC field campaign at similar scales. This transition shows the development of small shallow clouds into larger flower-type clouds with detrainment, triggered by increased mesoscale organization over several tens of kilometers. Beucher et al. (2022) utilized the French convection-permitting model AROME-OM at kilometer scales, successfully simulating four primary mesoscale patterns observed during the EUREC4A campaign. Despite the success of simulating the realism of the mesoscale structure of marine stratocumulus, further refinement may continue to be needed to enhance connections between radiation, microphysics, and planetary boundary layer schemes for adequately simulating the complexity of aerosol-cloud interactions.

In all 10 case studies, LWP adjustments were positive. This result remained consistent even when different PBL and microphysics schemes were employed. Despite the diversity in meteorological conditions, we were unable to simulate the negative LWP responses sometimes reported in LES studies, albeit using different boundary conditions (Ackerman et al., 2004; Seifert et al., 2015). Negative LWP responses have been documented in satellite observations of ship tracks (Christensen and Stephens, 2012), downstream from volcanic aerosol emissions (Malavelle et al., 2017; Toll et al., 2017), and more broadly in non-precipitating clouds, particularly in the presence of excessive dry air conditions above the marine boundary layer (Chen et al., 2014). Our findings generally align with positive LWP responses also identified in WRF LES simulations from the same region used in Wang et al. (2020). Although, the absence of a negative LWP response in our study may be attributed to a variety of processes. First, uncertainties in the autoconversion rate (a tunable parameter that affects the formation rate of raindrops) may lead to a positive LWP response as droplet number concentrations increase if this rate is underestimated (Mülmenstädt et al., 2020; Christensen et al., 2023). Second, if the sedimentation and entrainment rates are not strong enough in the model, the entrainment of overlying air may not be effective at the removal of cloud and rainwater (Bretherton et al., 2007).

While the MYNN3 PBL scheme parameterizes entrainment mixing reasonably well in the gray-zone (Ching et al., 2014), poorly resolved sub-kilometer scales can result in weaker increases in liquid water path with aerosols due to fewer precipitating

clouds and weaker LWP increase in non-raining clouds (Terai et al., 2020) within multi-scale climate models. Generally, these km-scale resolutions are well-suited for resolving the cumulus outflow, but they may still be too course to resolve updrafts well (Atlas et al., 2022). Ghonima et al. (2017) evaluated the MYNN scheme and other turbulence parameterization schemes using single-column model experiments showing that entrainment flux tendencies in stratocumulus tend to be underestimated compared to LES, resulting in cooler, moister stratocumulus-topped boundary layers. This discrepancy may imply a deficiency in representing strong turbulent mixing near the cloud top in our simulations. However, our simulations show an enhanced peak in the resolved TKE near the top of the stratocumulus cloud layer (Figure 9i). Also, when radiation is deactivated, TKE is much smaller and the cloud layer becomes significantly shallower (Figure S10), highlighting the role of radiative processes in driving stronger TKE throughout the boundary layer. WRF version 4.2 introduced scale-awareness, dynamically adjusting parameterized turbulent kinetic energy as resolution decreases, thus offering a more explicit representation of turbulent processes at finer scales (Olson et al., 2019). Subgrid-scale clouds produced by the MYNN-EDMF (section 3) are coupled to the longwave and shortwave radiation schemes (namelist parameter icloud_bl is set to 1). Despite these couplings, uncertainties may persist due to relatively coarse vertical resolution (compared to LES) and the ability to capture nonlocal production of TKE associated with cloud-top radiative cooling. Alternative approaches, such as explicit entrainment or employing the mass-flux method for downdrafts, may offer improved parameterization of destabilized parcels in stratocumulus environments (Olson et al., 2019). The impacts of model caveats like these on cloud cell expansion due to increased aerosol concentration should be explored in subsequent research with higher resolution models including LES where the cloud-top entrainment interface can be modeled at finer spatial scale resolutions. Nevertheless, our model set up shows evidence that radiative cooling drives stronger turbulence in the marine boundary layer but it remains crucial to constrain such parameters based on observations (Suzuki and Stephens, 2009; Golaz et al., 2013; Christensen et al., 2023; Varble et al., 2023), where possible, to enhance model development and our understanding of aerosol-cloud interactions and radiative forcing.

Overall, these WRF simulations suggest that an increase in aerosol concentration may result in significantly more radiative cooling than would otherwise be predicted by the Twomey effect at the relatively short spatio-temporal-scales (300 km over 12 hours) considered here. We find generally that aerosols expand the area of stratocumulus cells, increase liquid water path, and cloud fraction. These relationships become enhanced in the presence of precipitation. Given the link between these radiative impacts and the nature of the mesoscale organization of clouds and its sensitivity to aerosol, it may be prudent to resolve these radiative effects in larger-scale climate models for improved assessments of climate change.

*Code and data availability.* All ARM and ACE-ENA products are available at https://www.arm.gov/data/. CERES SYN Ed4.1 product is available at https://ceres.larc.nasa.gov. MODIS collection 6 products are available at https://earthdata.nasa.gov. MERRA-2 data were obtained from https://goldsmr4.gesdisc.eosdis.nasa.gov/data/MERRA-2/. HYSPLIT trajectory code is available at https://www.ready.noaa.gov/HYSPLIT.php. An archive of the WRF namelist.input and trajectory files for each case study day are provided at https://portal.nersc.gov/project/m1657/wrf_lagrangian_aci/ All data and code availability websites were last accessed on 10 October 2023.

*Video supplement.* Movies S1 related to this article is available in the supplementary materials.

*Author contributions.* MWC wrote the manuscript and developed the Lagrangian trajectory approach and analysis. PW guided the implementation of the cloud segmentation algorithm. Research and development ideas, as well as writing and editing, were contributed by PW, ACV, HX, and JDF.

*Competing interests.* At least one of the (co-)authors is a member of the editorial board of Atmospheric Chemistry and Physics.

*Acknowledgements.* We would like to thank the reviewers, Michael Diamond and an anonymous reviewer, as well as to the editor, Tim Garrett, for comments that improved the manuscript. We would also like to thank Yuwei Zhang for valuable feedback and assistance in compiling and running the WRF model. This research has been supported by the Atmospheric System Research (ASR) program as part of the US Department of Energy, Office of Science, Office of Biological and Environmental Research under Pacific Northwest National Laboratory (PNNL) project 57131. PNNL is operated for the US Department of Energy by Battelle Memorial Institute under contract DE-A06-76RLO 1830. Observations from the ENA site and ACE-ENA campaign are supported by the Atmospheric Radiation Measurement (ARM) Climate Research Facility.

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

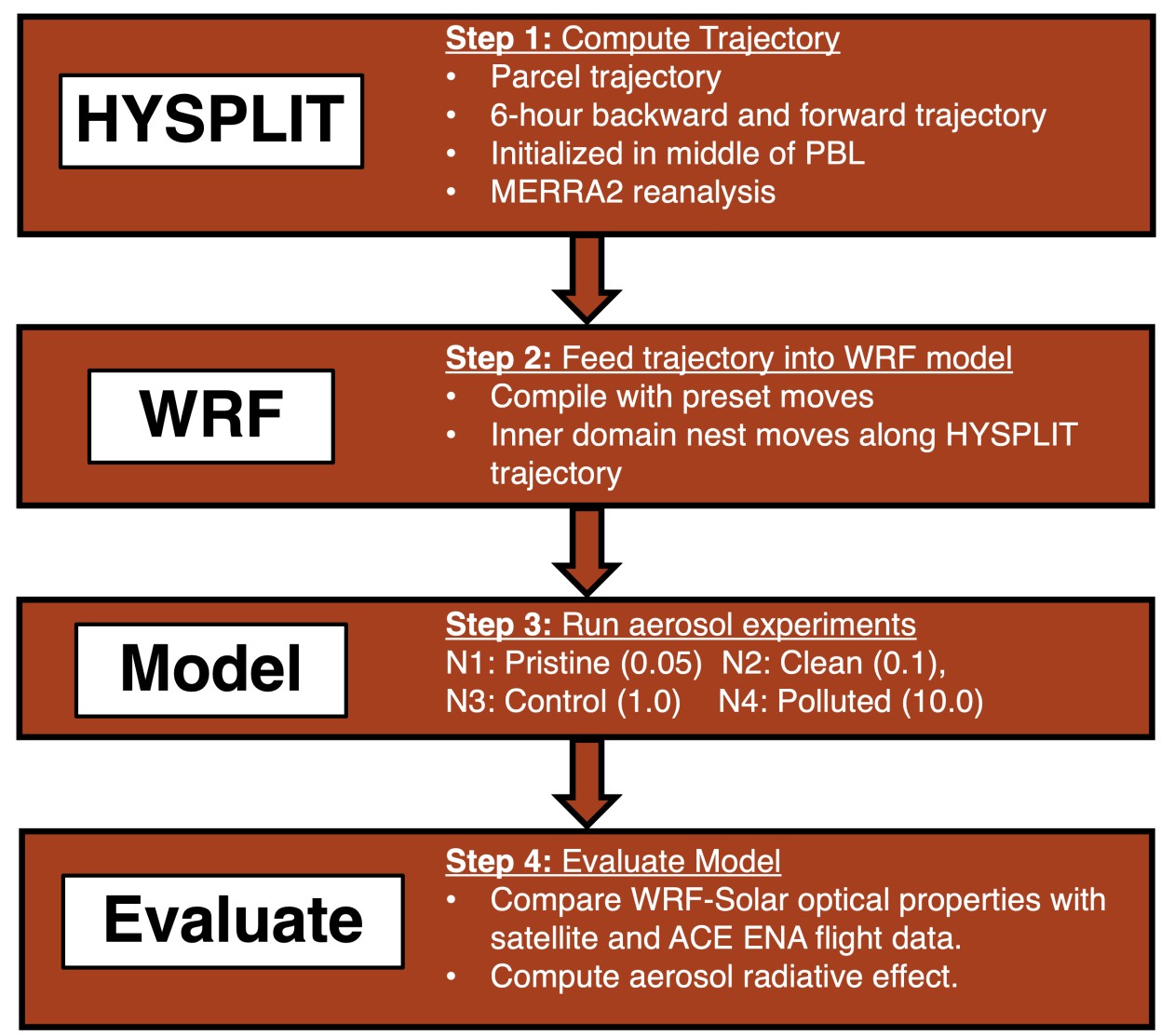

**Figure 1.** Flow chart depicting the methodology for studying aerosol-cloud interactions in a Lagrangian framework using the WRF model.

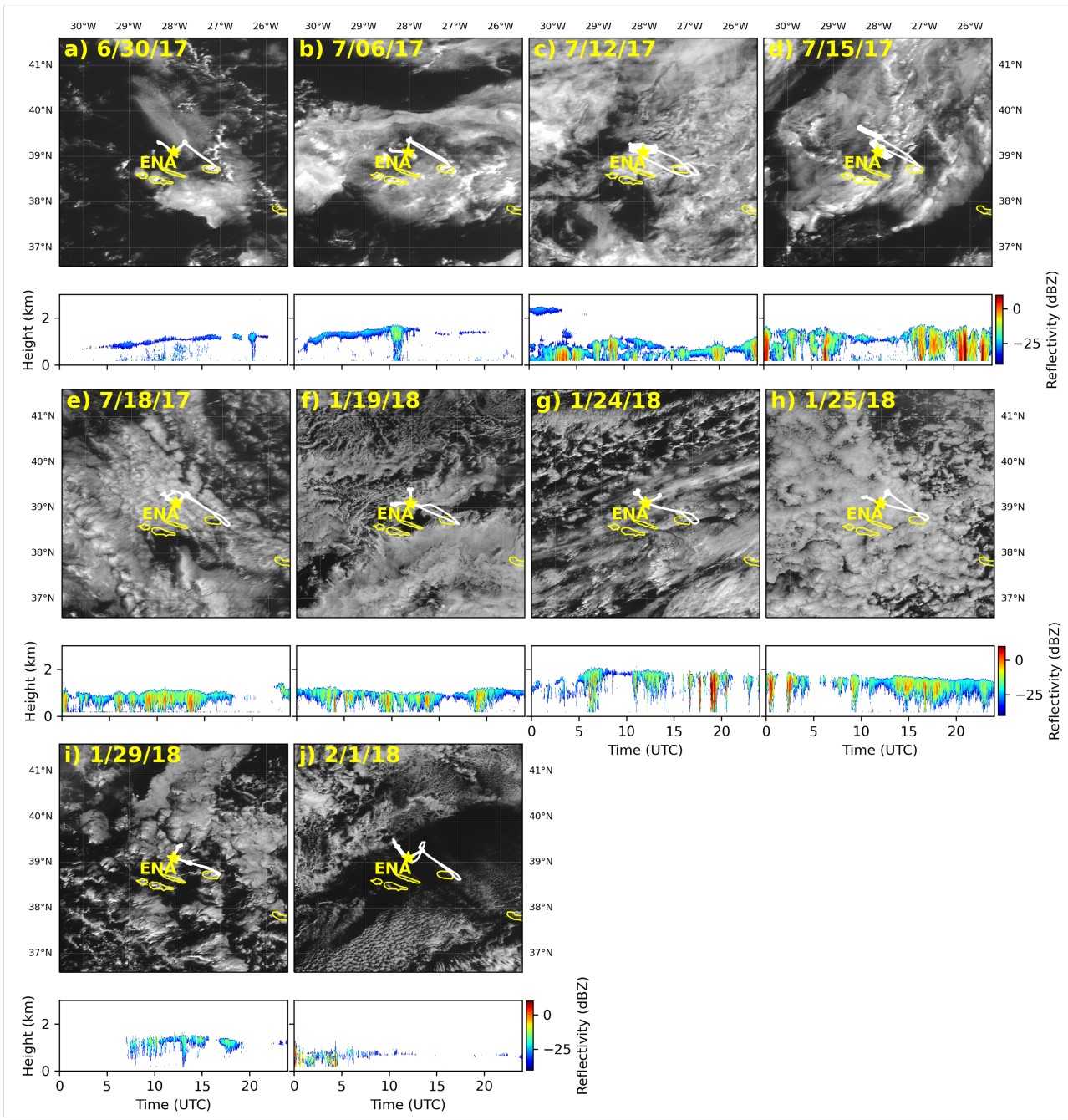

**Figure 2.** Case studies during summer (a - e) and winter (f - j) ACE-ENA IOP periods. Panels show GOES visible images at 13:00 UTC displayed over 4 × 4 ° regions centered over Graciosa Island (yellow star denotes the ARM site). Aircraft flight positions are shown as white lines. Vertically pointing Ka-band reflectivity at the ARM site is displayed over a 24-hr period for the corresponding days.

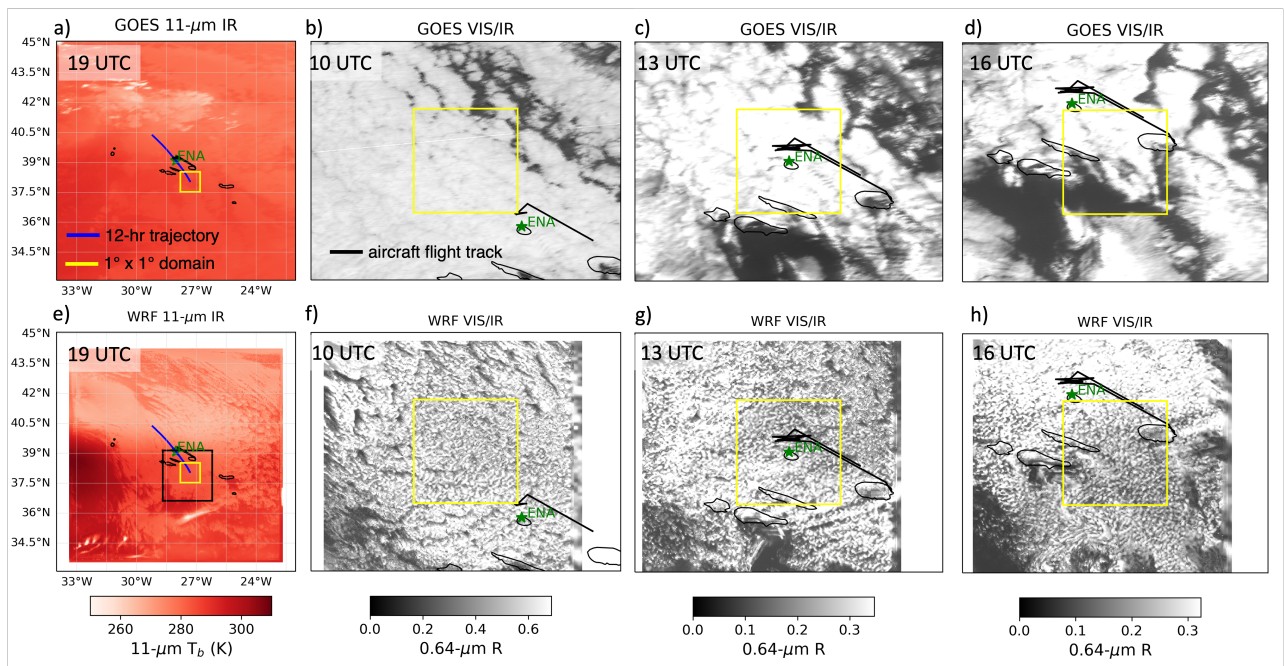

**Figure 3.** GOES 11-$\mu$m thermal infrared image on 07/18/2017 at 19:00 UTC is centered over Graciosa island with positions of the HYSPLIT trajectory computed using ERA5 reanalysis (blue line) a). The yellow box denotes a $1 \times 1°$ region that moves along the center of the trajectory. Visible imagery at 0.64-$\mu$m reflectance over a larger $2 \times 2°$ region is shown at discrete times (10, 13, and 16 UTC; b, c, d respectively). WRF simulations of the brightness temperature at 11 $\mu$m and normalized shortwave albedo are displayed at the same times (e, f, g, and h). The black line denotes aircraft observations from the ACE-ENA campaign.

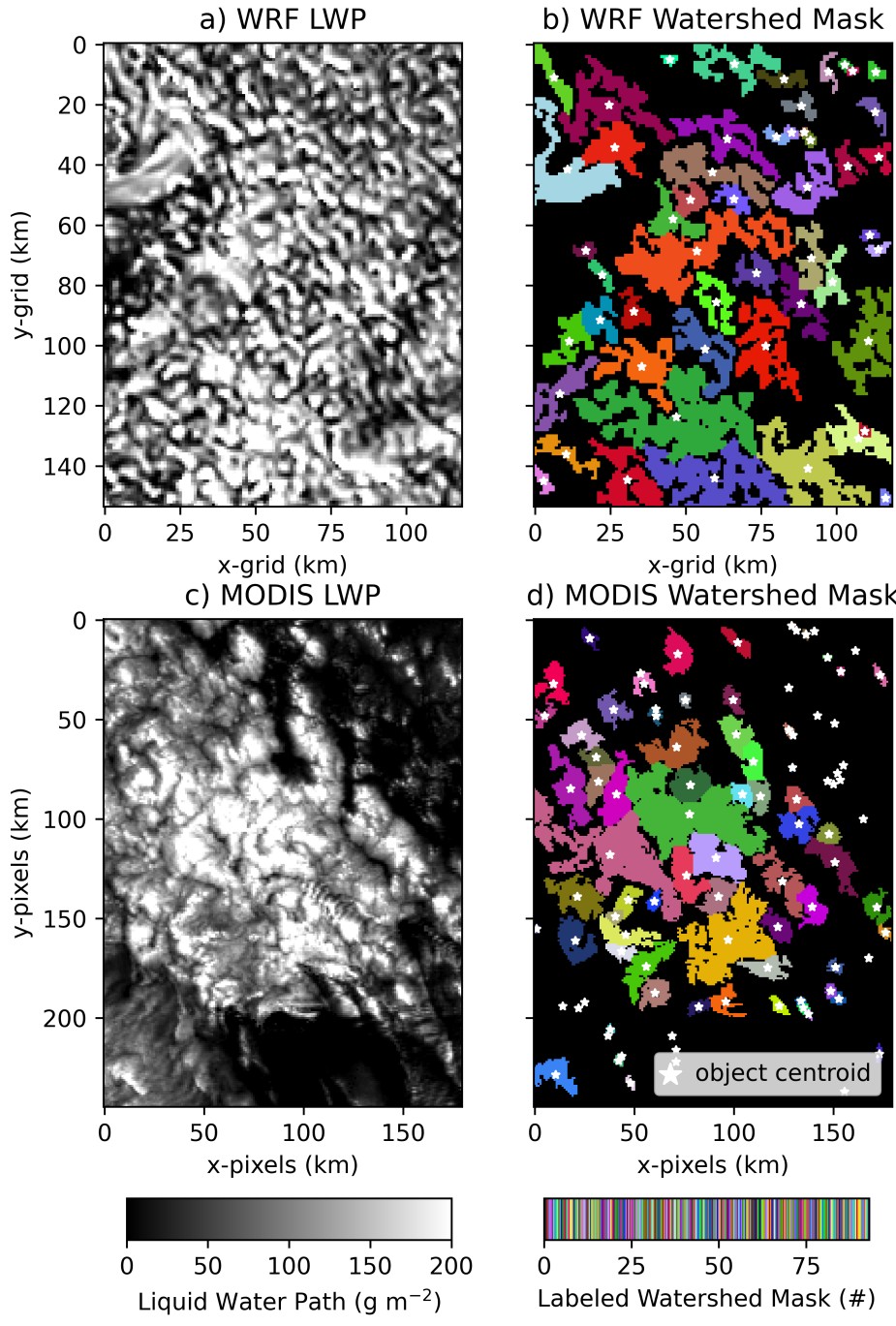

**Figure 4.** Liquid water path and watershed regions for WRF (a and b) and MODIS (c and d) on 07/18/2017 at 13 UTC. White stars indicate object centroid locations.

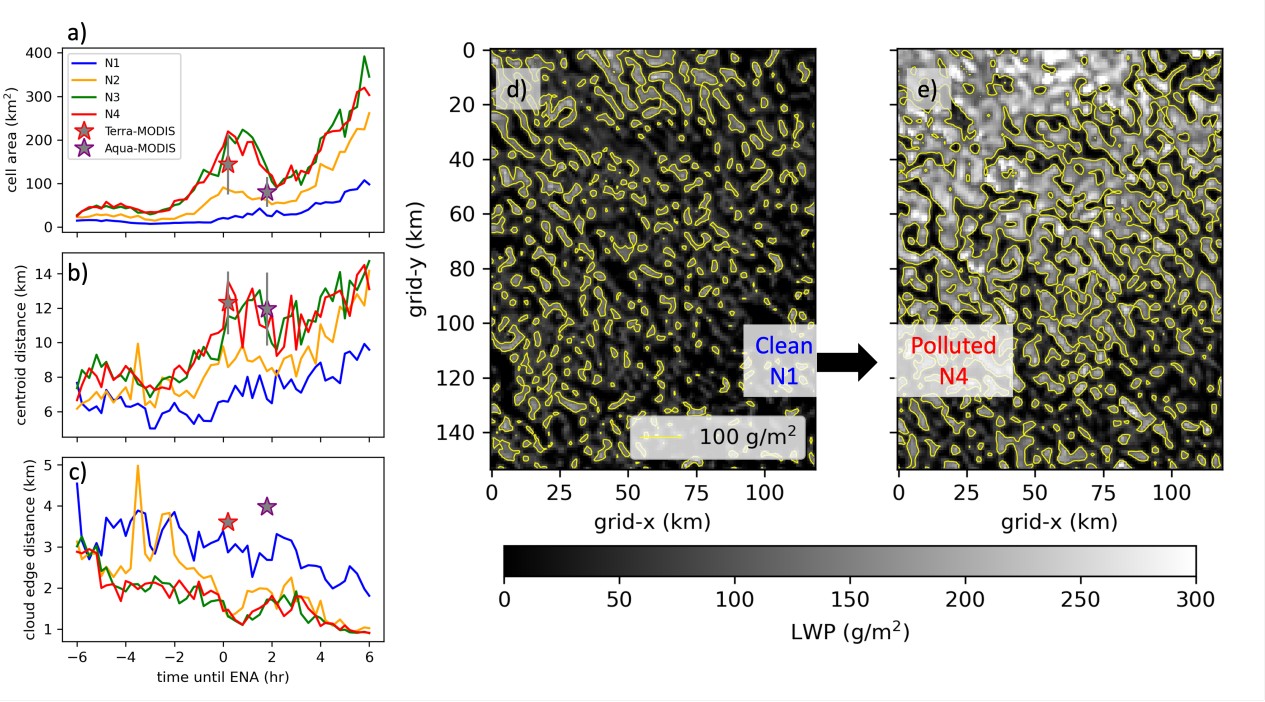

**Figure 5.** Time-series of the average (a) cloud object area, (b) minimum distance between cloud centroids, (c) minimum distance between cloud edges over each 15-minute time-interval detected for pristine (blue), clean (orange), control (green), and polluted (red) experiments on 07/18/2017. MODIS averages (star) and standard deviations (vertical lines) are displayed. Images of the LWP at 13 UTC is displayed for the clean (d) and polluted experiments (e).

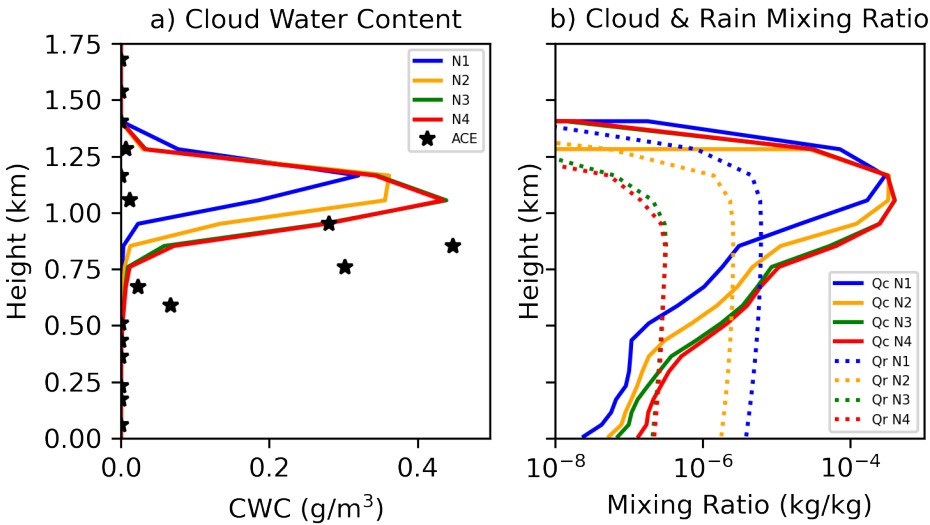

**Figure 6.** a) Vertical profile of the total water content measured by the G-1 aircraft in the WCM-2000 data product during the ACE-ENA flight (stars) on 07/18/2017 is averaged over an hour across the domain from 13-14 UTC.The flight path is illustrated in Figure 3 and simulated for pristine (N1; blue), clean (N2; orange), control (N3; green), and polluted (N4; red) experiments. Additionally, the simulated vertical profile of cloud (solid) and rain (dotted) water mixing ratios is averaged over the domain for each aerosol experiment.

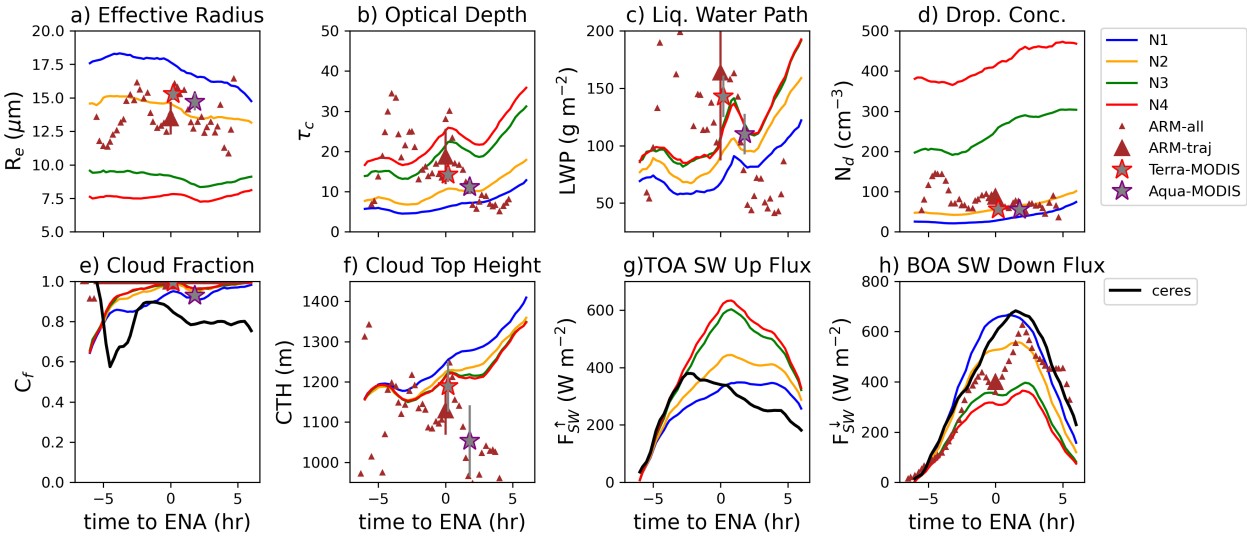

**Figure 7.** a) Droplet effective radius ($R_e$), b) cloud optical thickness ($\tau_c$) retrieved from the 3.7-$\mu$m band, c) liquid water path (LWP), d) droplet concentration ($N_d$) computed from $R_e$ and $\tau_c$, e) liquid cloud fraction ($C_f$), f) cloud top height (CTH), g) top of atmosphere outgoing shortwave radiative flux ($F_{SW}^{\uparrow}$, and h) bottom of atmosphere incoming shortwave flux ($F_{SW}^{\downarrow}$ for pristine (blue), clean (orange), control (green), and polluted (red) WRF simulations. WRF-Solar was used for comparison with the satellite retrievals. ARM (brown diamond) retrievals are provided at all time steps and at the time when the trajectory passes over the ARM site (larger brown diamond) and MODIS retrievals from satellites Terra (red star) and Aqua (blue star) are provided when available along the trajectory on 07/18/2017. Hourly retrievals of the cloud fraction and radiative fluxes are provided by CERES. Note, aside from time to ENA equals 0, the ARM measurements do not coincide with the trajectory location and are merely used to show Eulerian variability.

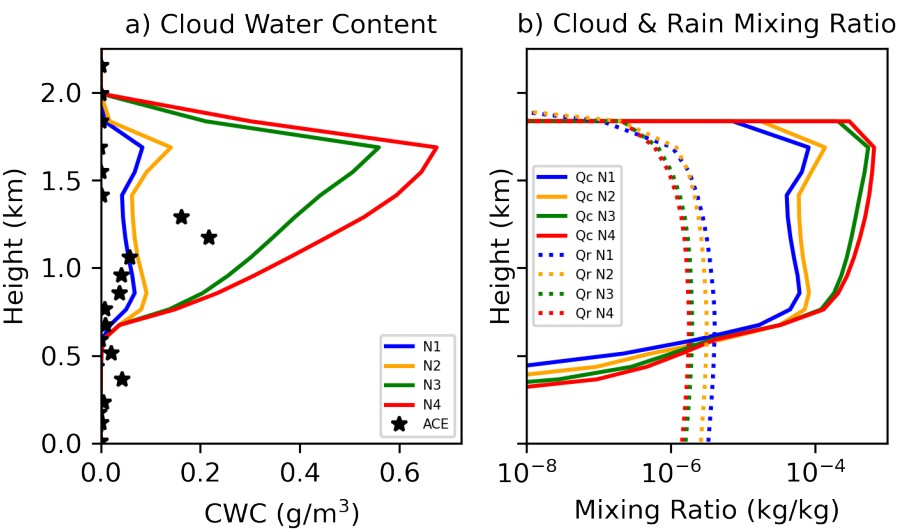

**Figure 8.** Same as Figure 6 except for case study 07/15/2017.

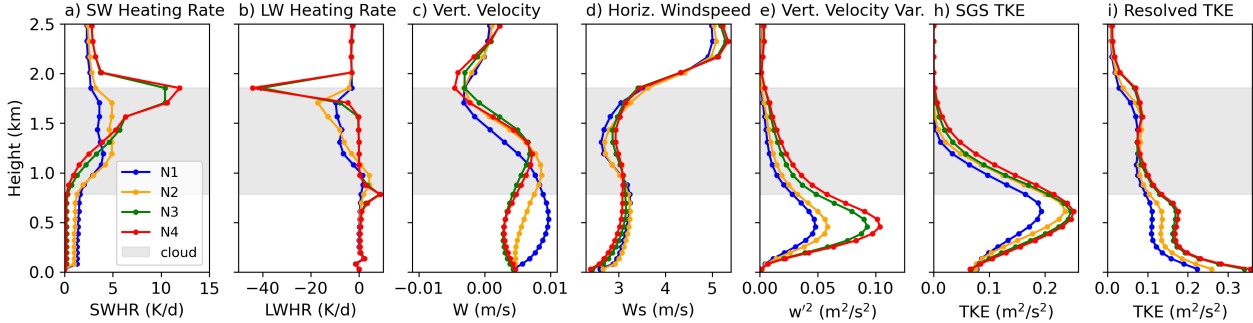

**Figure 9.** Vertical profile of the a) mean shortwave and b) longwave radiative heating rate (SWHR and LWHR), c) mean vertical velocity (W), d) mean horizontal wind speed (Ws), e) vertical velocity variance ($w'^2$), h) subgrid scale (SGS) turbulent kinetic energy (TKE) from MYNN, and i) resolved TKE computed from the 3D wind variances calculated from 3.2x3.2 km$^2$ regions averaged over the domain for pristine (blue), unpolluted (orange), control (green), and polluted (red) WRF simulations on 07/15/2017 at 13:00 UTC. Gray shading indicates the boundaries of the cloud layer for the control experiment.

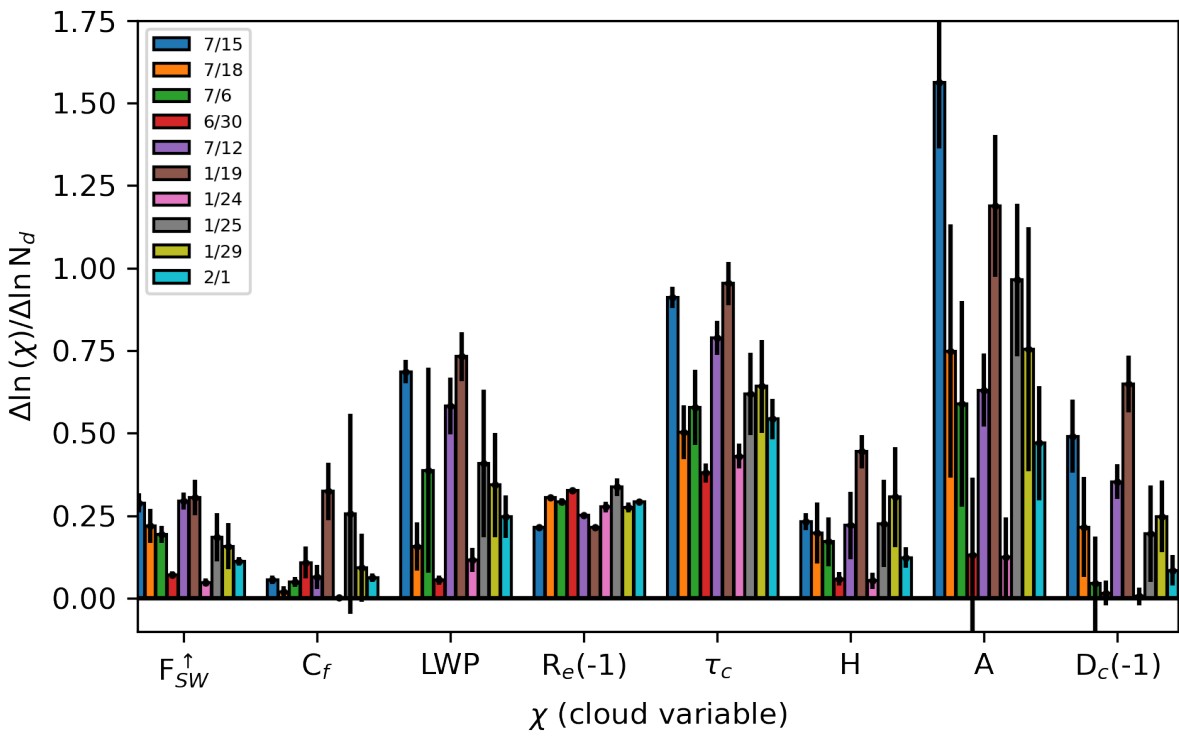

**Figure 10.** Value of the slope in the natural log change of a given variable ($\chi$) with respect to the natural log change in cloud droplet number concentration ($N_d$) computed from 4 aerosol WRF experiments in 10 different case studies (7/15, 7/18, 7/6, 6/30, 7/12, 1/19, 1/24, 1/25) represented at 13:00 UTC. $\chi$ variables shown are the top of atmosphere outgoing shortwave flux ($F_{SW}^{\uparrow}$), liquid cloud fraction ($C_f$), liquid water path (LWP), effective droplet radius ($R_e$), cloud optical thickness ($\tau_c$), cloud geometrical thickness (H), cloud object area extent ($A$), and distance between cloud object centroids ($D_c$). Multiplication of -1 on $R_e$ and $D_c$ was carried out to make all quantities positive across the bar chart. Uncertainties are represented by the 1-sigma error of the regression fit between quantities.

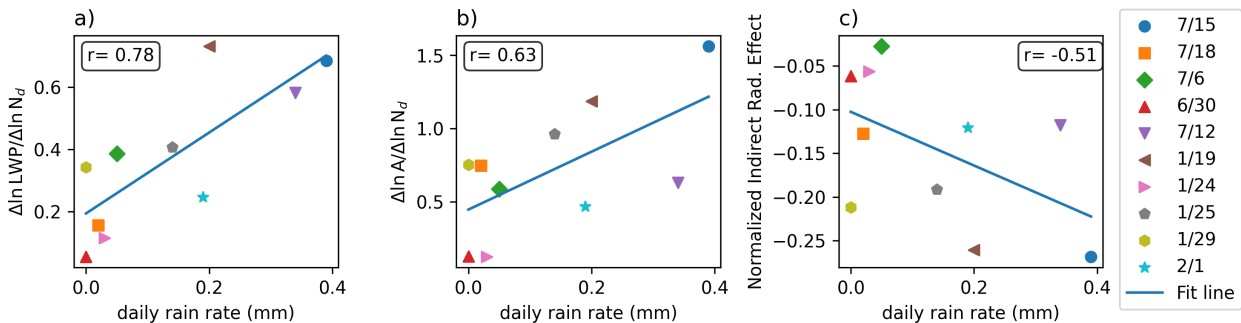

**Figure 11.** Scatter plot of the a) change in liquid water path ($\frac{\Delta \ln L}{\Delta \ln N_d}$), b) change in cellular cloud area as a function of $N_d$, and c) normalized indirect radiative effect which constitutes the Twomey + $\text{LWP}_{adj.}$ + $\text{CF}_{adj.}$ as a function of daily accumulated rain rate from ARM for simulations $\pm$ 3 hours from the time the trajectory intersects Graciosa Island for each case study day, designated by a different symbol as shown in the legend.

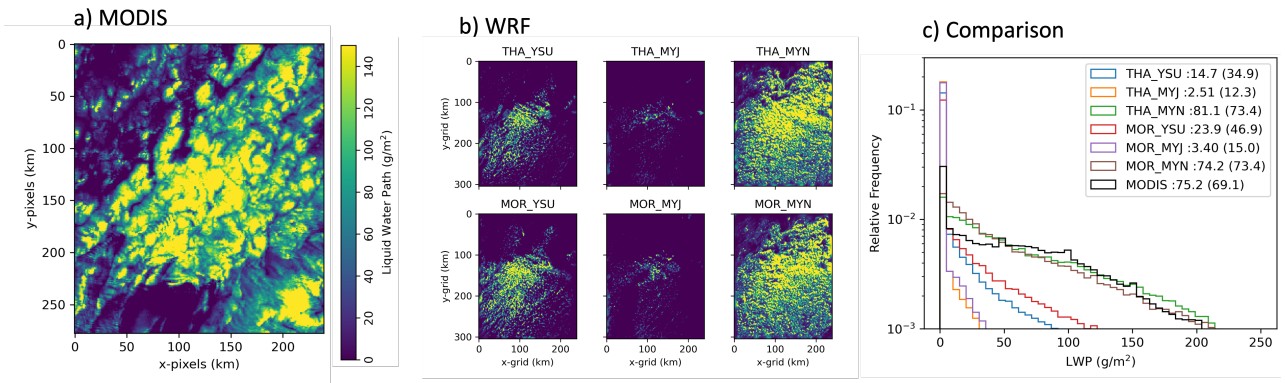

**Figure 12.** WRF control experiment using combinations of the Thompson (THA), Morrison (MOR), Yonsei University (YSU), Mellor–Yamada–Janjic (MYJ), and Mellor-Yamada-Nakanishi-Niino (MYN) boundary layer schemes. Spatial distributions of liquid water path (LWP) is shown for (a) MODIS retrieved using the 3.7-$\mu$m channel at 14:40 UTC, (b) WRF inner domain at 13:00 UTC, and (c) a histogram of LWP for experiment combination (c) with means and standard deviations displayed.

**Table 1.** WRF model schemes used to study aerosol-cloud interactions. Values for the coinciding names denote the option number used in WRF.

| WRF scheme | Value | Name |
|---|---|---|
| microphysics | 28 | Thompson (aerosol-aware) |
| radiation | 4 | RRTGM |
| cumulus | 6 | Tiedtke |
| pbl | 6 | MYNN |
| sfclay_physics | 2 | eta similarity |
| surface physics | 2 | Noah Land |

**Table 2.** Case studies from ACE-ENA IOP periods used to simulate stratocumulus clouds in WRF. Surface temperature ($T_s$), lower tropospheric static stability (LTS), free tropospheric entraining relative humidity at 850hPa (FTH), PBL height (determined from the temperature and humidity sounding), cloud base height (determined from the ceilometer), and daily integrated rainfall determined from ARM distrometer observations. Dominant cloud type following Wood and Hartmann (2006) classification based on satellite imagery inspection are listed.

| | $T_s$ [°C] | LTS [K] | FTH | PBL height [m] | Cloud base height [m] | Rainfall [mm] | Cloud type | Precipitation |
|---|---|---|---|---|---|---|---|---|
| **IOP 1** | | | | | | | | |
| 6/30/17 | 20.0 | 20.0 | 36 | 890 | 950 | 0 | disorganized | non-raining |
| 7/06/17 | 21.5 | 20.2 | 26 | 1410 | 1107 | 0.05 | homogeneous | light-rain |
| 7/12/17 | 22.0 | 17.2 | 72 | 1130 | 325 | 0.34 | homogeneous | moderate rain |
| 7/15/17 | 16.0 | 22.0 | 60 | 1530 | 850 | 3.9 | homogeneous | heavy-rain |
| 7/18/17 | 22.0 | 18.2 | 63 | 950 | 682 | 0.02 | closed-cells | non-raining with overlying cloud layers |
| **IOP 2** | | | | | | | | |
| 1/19/18 | 16.5 | 16.0 | 52 | 950 | 816 | 0.2 | open-cells | rain |
| 1/24/18 | 14.0 | 18.0 | 32 | 1710 | 1411 | 0.03 | open-cells | light-raining with ice |
| 1/25/18 | 13.0 | 19.7 | 21 | 1510 | 1302 | 0.14 | closed-cells | drizzle with ice |
| 1/29/18 | 15.0 | 18.1 | 50 | 1200 | 1062 | 0 | disorganized | non-raining |
| 2/01/18 | 15.0 | 17.8 | 41 | 600 | 565 | 0.19 | disorganized | drizzle |

**Table 3.** Twomey radiative effect, along with the liquid water path and cloud fraction adjustments relative to the Twomey effect (calculated using equation 1), are listed. Mean values across all case studies and excluding 1/25 due to excessive aerosol-induced cloud growth are included in the last two rows.

| Case | Twomey (W m$^{-2}$) | $\frac{LWP_{adj}}{Twomey}$ | $\frac{C_{fadj}}{Twomey}$ |
|---|---|---|---|
| 7/15/17 | -29.9 | 1.48 | 0.53 |
| 7/18/17 | -27.5 | 0.41 | 0.05 |
| 7/6/17 | -5.6 | 0.11 | 0.66 |
| 6/30/17 | -14.8 | 0.14 | 0.24 |
| 7/12/17 | -24.0 | 0.48 | 0.20 |
| 1/19/18 | -10.0 | 1.58 | 0.64 |
| 1/24/18 | -4.7 | 0.37 | 0.01 |
| 1/25/18 | -4.1 | 0.67 | 3.90 |
| 1/29/18 | -7.9 | 1.39 | 1.29 |
| 2/1/18 | -8.5 | 0.55 | 0.54 |
| Mean (all) | -13.7 ± 9.3 | 0.72 ± 0.53 | 0.81 ± 1.09 |
| Mean (excluding 1/25) | -14.7 ± 9.2 | 0.72 ± 0.56 | 0.46 ± 0.37 |

**Table 4.** WRF model setup for control and sensitivity experiments. Values in parenthesis denote the option number used in WRF. Experimental setup primarily used for analysis of detailed aerosol-cloud interaction experiments is listed in bold.

| Experiment name | Microphysics | PBL |
|---|---|---|
| THA_YSU | Thompson (28) | YSU (1) |
| THA_MYJ | Thompson (28) | MYJ (2) |
| **THA_MYN** | Thompson (28) | MYNN3 (6) |
| MOR_YSU | Morrison (10) | YSU (1) |
| MOR_MYJ | Morrison (10) | MYJ (2) |
| MOR_MYN | Morrison (10) | MYNN3 (6) |