# Peer review of "Aerosol-Induced Closure of Marine Cloud Cells: Enhanced Effects in the Presence of Precipitation"

_EGUsphere, 2023_

## Referee Comment (RC1)

This paper examines the impact of aerosols on the evolution of marine clouds and their cellular patterns by using the WRF model in a Lagrangian framework. Overall, the results from several cases are interesting, and the experimental approach helps understand the impact of aerosol on cloud evolutions. The results from several case study experiments in WRF show that increased aerosol concentration suppressed drizzle and increased cloud water content. These changes can lead to larger radiative cooling rates at cloud top because droplet size is smaller and concentration is larger in polluted clouds. Thus, the authors mentioned that the vertical and horizontal wind speeds near the base of the lower tropospheric inversion increase, making marine cloud cells larger and the gap between shallow clouds smaller. However, the connection between the main results is not clear, and the explanation is insufficient to support them. I think the authors already showed many figures in the main text and supplement to support the results. However, some work is needed to minimize confusion about this finding and its implications. The results will merit publication in ACP if the authors are able to address my concerns. I hope my comments below will clarify a few points about the results.

Main comments:

My primary concern is about the capability of the WRF model to represent the entrainment and mixing near cloud top-driven radiative and evaporative cooling due to its vertical resolution. In the manuscript, the authors conclude that increased aerosol concentration leads to larger radiative cooling rates and stronger wind shear near cloud top. These changes are closely related to the enhancement of entrainment and mixing of dry air above cloud top. As shown in Table 2, most cases show that free troposphere entraining relative humidities are low, which means the evaporation of cloud droplets is more efficient if entrainment-mixing is enhanced due to larger radiative cooling and stronger wind shear. Therefore, enhanced free-tropospheric and cloudy air mixing can decrease cloud water content and broaden the gap between the clouds. However, this would be inconsistent with the main results in this manuscript. I am quite concerned about whether this model can represent the effect of cloud top mixing driven by radiative and evaporative cooling because the vertical resolution of this model is too coarse, about 50 to 100 m near the cloud top. Do the results here imply that the effects of cloud top mixing were appropriately represented? I think more information is needed regarding the cloud-top mixing effects for the results. For example, for each aerosol case, you can show the vertical profiles of some variables related to entrainment-mixing (e.g., the entrainment rate and evaporative cooling rate). I strongly recommend that the authors revise the abstract and conclusions to reduce the emphasis and confidence level about the statements related to the model's inability for cloud-top processes. I believe it would minimize the confusion, as mentioned above.

Minor/grammatical comments:

-Figure 6: It seems that cloud water content is derived from the aircraft measurement dataset (FCDP+2DS+HVPS), correct? If so, it needs to be explained how to derive it in detail.

-Figure 7: The droplet number concentration from the measurements is close to N2 case, and the liquid water path is slightly larger than N3 and N4 cases. However, the effective radius is similar to N2. I am not sure if it is correct. It needs to explain how to calculate an effective radius in detail. The brief information about "ceres" should be included in the caption.

-Figure 9: I could not find a similar figure on 07/18/2017 in the supplement. It should be included in the main text or supplement. Fig. 9(d) shows a slight difference in horizontal wind speed between pristine, unpolluted, control, and polluted. Can such a slight difference redistribute the clouds (expansion of cloud cells)?

Line 35: change "proposed by (Rosenfeld et al., 2006)" to "proposed by Rosenfeld et al. (2006)"

Line 300-306: The same figure for 07/18/17 should be included in the main text or supplement as mentioned above. Why does the rainfall suppression make the updrafts weaker in the lower PBL?

Line 464: If the sedimentation and entrainment rates are underestimated, the authors should show them for each case. I think it is not difficult to show them from the simulations.

---

## Author Comment (AC1)

**Aerosol-Induced Closure of Marine Cloud Cells: Enhanced Effects in the Presence of Precipitation**

Matthew W. Christensen, Peng Wu, Adam C. Varble, Heng Xiao, and Jerome D. Fast

**Referee Comments**

Point-by-point responses in blue, additions to manuscript are ***bold & italicized***
(***Line numbers are associated with the revised manuscript***)
Figures specifically for the response are labeled starting with "R"
Figures that are added to both the response and revision are the same.

Dear Referees,

Thank you for your time and effort invested in our manuscript. We appreciate your fair and insightful evaluation of this work, and your comments have resulted in substantive changes to the manuscript, enhancing the connection between the main results and their interpretation. Specifically, the newly incorporated details encompass limitations and caveats associated with our configuration of the WRF model in representing cloud-top mixing processes. Additionally, supplementary tables now present radiative effect estimates for all case study days, and we have investigated the aerosol impact on cloud-segment updrafts. Furthermore, a detailed evaluation of the WRF GOCART aerosol profiles has been included. After implementing these changes, we believe the conclusions are now stronger, and the overall narrative remains the same.

Best regards,
Matt
* * *
**Referee 1 Comments**

This paper examines the impact of aerosols on the evolution of marine clouds and their cellular patterns by using the WRF model in a Lagrangian framework. Overall, the results from several cases are interesting, and the experimental approach helps understand the impact of aerosol on cloud evolutions. The results from several case study experiments in WRF show that increased aerosol concentration suppressed drizzle and increased cloud water content. These changes can lead to larger radiative cooling rates at cloud top because droplet size is smaller and concentration is larger in polluted clouds. Thus, the authors mentioned that the vertical and horizontal wind speeds near the base of the lower tropospheric inversion increase, making marine cloud cells larger and the gap between shallow clouds smaller. However, the connection between the main results is not clear, and the explanation is insufficient to support them. I think the authors already showed many figures in the main text and supplement to support the results. However, some work is needed to minimize confusion about this finding and its implications. The results will merit publication in ACP if the authors are able to address my concerns. I hope my comments below will clarify a few points about the results.

**Main comments:**

My primary concern is about the capability of the WRF model to represent the entrainment and mixing near cloud top-driven radiative and evaporative cooling due to its vertical resolution.

>> Thank you for your insightful comments and for raising concerns regarding vertical mixing. Our horizontal grid spacing for the inner domain is 800 m, which is too coarse to resolve eddies responsible for stratocumulus-top entrainment mixing, regardless of how fine the vertical resolution is. We rely on the MYNN3 PBL scheme to parameterize most of the entrainment mixing. The MYNN3 PBL scheme has been shown to perform reasonably well in gray zone resolutions (see, e.g., Ching et al. 2014). The debate over how well these PBL schemes capture the complex interactions among radiation, microphysics, and turbulence in the entrainment zone is ongoing. Even Large Eddy Simulations (LES) of stratocumulus-topped Planetary Boundary Layers (PBL) show strong sensitivity to their subgrid scale (SGS) parameterizations (Mellado et al. 2018).

References
Ching, J., Rotunno, R., LeMone, M., Martilli, A., Kosovic, B., Jimenez, P. A., and Dudhia, J.: Convectively Induced Secondary Circulations in Fine-Grid Mesoscale Numerical Weather Prediction Models, Monthly Weather Review, 142, 3284 – 3302, https://doi.org/10.1175/MWR-D-13-00318.1, 2014

Mellado, J. P., Bretherton, C. S., Stevens, B., & Wyant, M. C. (2018). DNS and LES for simulating stratocumulus: Better together. *Journal of Advances in Modeling Earth Systems*, 10, 1421–1438. https://doi.org/10.1029/2018MS001312

In the manuscript, the authors conclude that increased aerosol concentration leads to larger radiative cooling rates and stronger wind shear near cloud top. These changes are closely related to the enhancement of entrainment and mixing of dry air above cloud top. As shown in Table 2, most cases show that free troposphere entraining relative humidities are low, which means the evaporation of cloud droplets is more efficient if entrainment-mixing is enhanced due to larger radiative cooling and stronger wind shear. Therefore, enhanced free-tropospheric and cloudy air mixing can decrease cloud water content and broaden the gap between the clouds. However, this would be inconsistent with the main results in this manuscript.

>> First, the stratocumulus-top entrainment is self-limiting in the sense that if it "decreases cloud water content and broadens the gap between the clouds (decreasing cloud cover)," the radiative/evaporative cooling also decreases, and the entrainment mixing it induces, in turn, decreases. In other words, if entrainment drying is so desiccating to the cloud layer, the cloud layer would become thinner, and the PBL would then adjust to a state with reduced entrainment, leading to a shallower PBL if other factors (e.g., subsidence) remained unchanged. We do not observe a shallower PBL in the polluted case of 7/15/17. On the other hand, an increase in cloud-top buoyancy production, whether through enhancements in radiative or evaporative cooling, not only intensifies entrainment mixing near the cloud top but also results in stronger overall TKE and moisture transport from the surface to the cloud layer (unless the cloud layer is decoupled from the subcloud layer). This, in turn, generates more clouds as the PBL deepens due to enhanced entrainment mixing. We observe increases in both cloud LWP and PBL/cloud-top heights in the polluted case for case study 7/15/17. For the 7/18/17 case, the unpolluted and polluted states have similar mean PBL heights but the clean state fluctuates more due to more significantly resolved w'^2, indicating more resolved secondary/meso-scale circulation, possibly driven by the larger rainfall.

I am quite concerned about whether this model can represent the effect of cloud top mixing driven by radiative and evaporative cooling because the vertical resolution of this model is too coarse, about 50 to 100 m near the cloud top. Do the results here imply that the effects of cloud top mixing were appropriately represented? I think more information is needed regarding the cloud-top mixing effects for the results. For example, for each aerosol case, you can show the vertical profiles of some variables related to entrainment-mixing (e.g., the entrainment rate and evaporative cooling rate).

>> Unfortunately, we lack sufficient output to estimate the entrainment rate directly. However, as mentioned earlier, the increased cloud top/PBL height in the polluted case suggests that our simulations do capture, to some extent, the enhancement of entrainment (resulting in a deeper PBL) induced by enhanced cloud-top cooling. This is consistent with our expectation that domain-scale subsidence changes little between cases with different aerosol concentrations. In Figure R1 we present the TKE averaged from WRF columns across the same domain that are cloud-free and those that have thin, medium, and thick clouds, as determined by the LWP threshold. This result suggests that cloud-top buoyancy production by increased radiative cooling is driving TKE because the red line (columns with large LWP) maximizes at a higher altitude than the others.

[Figure]

Figure R1. Vertical profile of the Turbulent Kinetic Energy (TKE) for columns in the WRF model with no cloud layers (clear-sky; blue), liquid water path (LWP) between 0 and 100 g m$^{-2}$ (orange), LWP between 100 and 250 g m$^{-2}$ (green), LWP greater than 250 g m$^{-2}$ (red) on 7/15/17 at 13:00 UTC. The mean PBL top is approximately 1500 m for these profiles.

I strongly recommend that the authors revise the abstract and conclusions to reduce the emphasis and confidence level about the statements related to the model's inability for cloud-top processes. I believe it would minimize the confusion, as mentioned above.

>> We have revised the abstract and conclusions pointing out the limitations in mesoscale cloud modeling of ACI. For example, we have added these points to the conclusions section:

**L486-496:** *Although, the absence of a negative LWP response in our study may be attributed to a variety of processes. First, uncertainties in the autoconversion rate (a tunable parameter that affects the formation rate of raindrops) may lead to a positive LWP response as droplet number concentrations increase if this rate is underestimated (Mülmenstädt et al., 2020; Christensen et al., 2023). Second, sedimentation and entrainment rates can affect the removal of cloud and rainwater (Bretherton et al., 2007). While the MYNN3 PBL scheme parameterizes entrainment mixing reasonably well in the gray-zone (Ching et al., 2014), resolving sub-kilometer scales can result in weaker increases in liquid water path with aerosols due to fewer precipitating clouds and weaker LWP increase in non-raining clouds (Terai et al., 2020) within multi-scale climate models. Generally, these km-scale resolutions are well-suited for resolving the cumulus outflow, but they may still be too course to resolve updrafts well (Atlas et al. 2022). The impacts of model caveats like these on cloud cell expansion due to increased aerosol concentration should be explored in subsequent research with higher resolution models including large eddy simulations where the cloud-top entrainment interface can be modeled at finer spatial scale resolutions.*

We have also added this sentence to the abstract, **L13-L15:** *While higher resolution large eddy simulations may provide improved representation of cloud-top mixing processes, these results emphasize the importance of addressing mesoscale cloud-state transitions in the quantification of aerosol radiative forcing that cannot be attained from traditional climate models.*

**Minor/grammatical comments:**
-Figure 6: It seems that cloud water content is derived from the aircraft measurement dataset (FCDP+2DS+HVPS), correct? If so, it needs to be explained how to derive it in detail.
>> We have added "***total water content, measured by the G-1 aircraft in the WCM-2000 data product***" caption of Figure 6 as well as add more explanation to this dataset in section 2.2 of the text.
**L106-111:** *The multi-element water content measuring system utilizes a scoop-shaped sensor to measure total water content, capturing both liquid and ice phase hydrometeors. It incorporates two heated wire elements (021-wire and 083-wire), exposed directly to the airstream, along with a reference element exposed to the airflow but not to condensed water. Following the approach of Miller et al. (2022), we adopt the WCM-2000 system due to its favorable agreement in liquid water content measurements compared to the Fast Cloud Droplet Probe and Two-Dimensional Stereo particle imaging probe measurement systems*.

-Figure 7: The droplet number concentration from the measurements is close to N2 case, and the liquid water path is slightly larger than N3 and N4 cases. However, the effective radius is similar to N2. I am not sure if it is correct. It needs to explain how to calculate an effective radius in detail. The brief information about "ceres" should be included in the caption.
>> The revised manuscript now provides additional details regarding the effective radius retrieval in the caption and main body of the text. Specifically, we clarify that the effective radius used in the comparison is ***retrieved at 3.7-μm*** and that liquid water path and droplet number concentration are computed from effective radius and optical thickness (retrieved at 3.7-μm) in the caption. Furthermore, we add the following to section 2.3:

**L123-126:** *Of the three spectral channels used for $R_e$ retrievals, the sensitivity of the 3.7-µm channel is weighted closest to the cloud top, primarily due to the relatively strong absorption of water vapor at this wavelength (Platnick 2000). Because errors in the adiabatic droplet number concentrations using the 3.7-µm channel are considerably smaller than in the other bands (Grosvenor et al. 2018), we choose to use it for this study.*

CERES information has also been added to the caption.

Regarding the comparison, the close correspondence between effective radius (being close to the N2 line) and the cloud droplet number concentration (being close to N2 line) is expected due to the strong dependence (to the power of -2.5) of the effective radius on the droplet number concentration calculation (i.e. $N_d = \gamma\, \tau^{0.5} R_e^{-2.5}$). The comparison of optical depth and liquid water path (i.e. $LWP \sim \tau R_e$) is less by comparison due to its weaker dependence (to the power of 0.5).

-Figure 9: I could not find a similar figure on 07/18/2017 in the supplement. It should be included in the main text or supplement. Fig. 9(d) shows a slight difference in horizontal wind speed between pristine, unpolluted, control, and polluted. Can such a slight difference redistribute the clouds (expansion of cloud cells)?
>> Thank you for raising this point. We have added the corresponding figure to the supplement describing the radiative flux, wind, and turbulence profiles of the for the 07/18/2017 case study. It is also included below. **L320-321:** *Vertical profile shapes of these quantities are similar, albeit less pronounced, on 7/18 (Figure S9).* Regarding your last question, we wouldn't necessarily assume a direct relationship between horizontal/vertical wind speed and cloud expansion, as many other factors (as stated in the manuscript), such as radiative cooling rate, TKE, PBL depth, etc., could also influence the cloud object area. Nevertheless, we can simply estimate what the expansion rate would be based solely on the horizontal winds. The horizontal wind speed difference between the pristine (N1) and polluted case (N4) is ~0.5 m/s at its peak in the vertical profile near 1.3 km above the surface in Figure 9. This difference would lead to a change of ~10 km, assuming a constant rate over a 6-hour period. This value is nearly twice the centroid spreading of ~5 km over the same period (Figure 5). Thus, the horizontal wind speed differences are indeed large enough to redistribute the clouds (expansion of cloud cells), but we would prefer not to speculate that this variable is solely responsible for the cloud-cell expansion.

[Figure]

*Figure S9. Vertical profile of the a) longwave radiative cooling rate, b) turbulent kinetic energy, c) cloud water mixing ratio, and d) rain water mixing ratio for the control, no evaporative cooling from cloud and rain drops, no radiation to cloud layer, and turning off the cumulus scheme from the WRF experiments for the case study day 07/15/2017 at 13:00 UTC.*

Line 35: change "proposed by (Rosenfeld et al., 2006)" to "proposed by Rosenfeld et al. (2006)"
>> Done

Line 300-306: The same figure for 07/18/17 should be included in the main text or supplement as mentioned above. Why does the rainfall suppression make the updrafts weaker in the lower PBL?
>> We added a similar figure (see above; *Figure S9*) to the supplement for the 07/18/2017 case study and removed the speculative statement that "rainfall suppression" makes the updrafts weaker.

Line 464: If the sedimentation and entrainment rates are underestimated, the authors should show them for each case. I think it is not difficult to show them from the simulations.
>> The word "underestimated" was meant to be speculation rather than a direct comparison to observations of sedimentation and entrainment rates that are not available. To avoid confusion, this discussion now uses the word "uncertainties" in describing process representation in the model before discussing how they could affect LWP.

**Referee 2 Comments**

The authors set up Lagrangian nested WRF simulations at convection-permitting resolution for 10 cases of stratocumulus cloud evolution based on the availability of ACE-ENA flight data and find that as they increase aerosol concentration within the simulations, closed cellular cloud structures tend to expand horizontally (and somewhat vertically as well). The resulting adjustments enhancing liquid water path and cloud fraction together more than double the cooling that would result from the Twomey effect alone on average. Overall, the analysis is well done and the paper is interesting and well-written. I believe some additional nuance would be useful, however, particularly clarifying that the adjustments found in the work are not based on the observations and acknowledging the continuing limitations of the horizontal and vertical resolutions. The discussion of the cloud object method and interpretation could also be clarified. I recommend accepting the manuscript pending minor revisions. -MD

**General comments:**

**A) Model versus observational results:** The discussion should better clarify that all aerosol effect conclusions are based on model experiments only. There is no attempt made to deduce aerosol relationships from the observations themselves.

>> In the abstract and conclusions, we make a stronger point that the radiative effects are based on kilometer-scale model simulations (e.g. **L11-12, L52**) and the observations are used to validate (e.g. **L5**) the model. We have also emphasized when our *modeling* comparisons have been contrasted with *observational* estimates to make this distinction clearer throughout (e.g. **L475 – 478**).

**B) Resolution:** I agree with the comments about the vertical resolution mentioned by reviewer 1 and think this context should be emphasized more when discussing the positive LWP adjustments. The inability of models to properly represent entrainment and thus the mechanism believed to be behind observed negative LWP adjustments in pollution tracks and effusive volcanic plumes (e.g., Malavelle et al. 2017, Toll et al. 2017) has been repeatedly flagged, as the authors know well. I also think the discussion of horizontal resolution could use a bit more nuance, as the km-scale resolution is well-suited for resolving the cumulus outflow but is still too course to resolve the updrafts well (Atlas et al., 2022, have a nice treatment of this issue, for example).

>> Please see our response to reviewer 1. As you both suggest, we have added these very important limitations and caveats to the manuscript and describe the nuances in more detail in the conclusions section as well as in the abstract.

Malavelle, F. F., et al.: Strong constraints on aerosol-cloud interactions from volcanic eruptions, Nature, 546, 485-491, 10.1038/nature22974, 2017.

Toll, V., Christensen, M., Gassó, S., and Bellouin, N.: Volcano and Ship Tracks Indicate Excessive Aerosol-Induced Cloud Water Increases in a Climate Model, Geophysical Research Letters, 44, 12492– 12500, 10.1002/2017gl075280, 2017.

Atlas, R. L., Bretherton, C. S., Khairoutdinov, M. F., and Blossey, P. N.: Hallett-Mossop Rime Splintering Dims Cumulus Clouds Over the Southern Ocean: New Insight From Nudged Global

Storm-Resolving Simulations, AGU Advances, 3, e2021AV000454,
https://doi.org/10.1029/2021AV000454, 2022.
>> Thank you for the references. We have added them to appropriate locations in the manuscript.

**C) Cloud objects:** I'm having some trouble interpreting the cloud objects. It seems that for higher aerosol cases, separate updrafts with spreading anvils intersect with each other and are considered one cloud object whereas in the cleaner case they would be treated as separate objects. I can see how this might be helpful for thinking about overall cloud fraction, but it seems like it could be misleading if the number of distinct updrafts isn't changing between pollution cases.
>> Based on your comment, we have rigorously tested whether the number of distinct updrafts changes in cloud object segments between aerosol experiments. Below, Figure R2 shows the impact of aerosols on distinct updrafts occurring within cloud objects. Vertical velocity values are extracted from the cloud object area (e.g. for one object Figure R2a) from the surface to 2500 m, ranging from -2 to 2 m/s (Figure R2b). The number of updrafts with velocities greater than a $W_{threshold}$ is counted for each cloud object segment detected within the domain. $W_{threshold}$ ranges from 0 to 2.5 m/s in 25 bins. As previously shown, the average cloud segment area increases as aerosol loading increases (Figure R2c). Despite this increase, fewer relatively large updrafts (with w > 1.5 m/s) are found in polluted cloud objects (Figure R2d). Taking the ratio of object area to the number of updrafts shows that the cloud object areas are actually expanding for a given updraft (Figure R2e). These results are robust across a wide range of $W_{threshold}$, as shown in the line plot averages of the normalized area per number of updrafts as a function of $W_{threshold}$ (Figure R2f).

[Figure]

Figure R2. Cloud segment objects in WRF pristine (N1) simulations on 7/18/17 at 13 UTC with one segment example highlighted white with red pixel locations designating updraft locations a). A histogram of the vertical velocity of all grid-boxes

from the surface to 2500 m for this example segment b). Violin plot illustrating the distribution of the area values of all objects, the colored area indicating the data density with black lines of each violin representing the mean values and standard deviation c), number of updrafts greater than 1.5 m/s d), and number of updrafts determined from the 1.5 m/s threshold per unit object area e). Number of updrafts per unit object area averaged for all segments as a function of $W_{threshold}$ f).

To interpret these results, Figure R3 shows a diagram depicting two scenarios based on an assumed linear relationship between the area of the clouds and number of distinct updrafts (i.e. A = $\Delta A/\Delta n_u * n_u$) estimated from WRF simulations. When the *number of updrafts* is fixed, clouds become larger in area as aerosol increases (Scenario 1). When the *area* of the clouds is fixed, the number of updrafts decrease as aerosol increases (Scenario 2). Thus, fewer updrafts are needed to sustain the same size cloud under polluted conditions or larger cloud areas result from the same number of distinct updrafts under polluted conditions. Overall, the number of distinct updrafts in objects on average does change (decrease in this case) between aerosol simulations (Figure R2d). We hope this analysis better clarifies the connection between cloud object size, spreading anvils, and distinct updrafts.

[Figure]

Figure R3. Conceptual diagram showing the relationship between cloud area (square boxes) and number of distinct updrafts occurring between the surface and 2500 m for Scenario 1) where the number of updrafts and cloud area can change and Scenario 2) where the area of the cloud is fixed and distinct updrafts can change between pristine (blue) and polluted (red) simulations. Cloud expansion rate per unit updraft ($\gamma$) is obtained from WRF simulations displayed in Figure R2e.

**Specific comments:**
1. Lines 72-73: More explanation is needed for the "decreasing seasonal cycle" of CDNC result. I'm assuming you mean that aerosol concentrations are lower in winter than summer, but the higher activation fraction winter leads to a suppressed seasonal difference in CDNC?
>> We have removed "thereby resulting in decreasing the seasonal cycle in cloud droplet number concentration" and added the following sentence for clarity, **L73-75: *Despite higher***

*activated aerosol fractions in winter, droplet number concentrations are lower due to less available aerosol compared to summer conditions (Wang et al. 2022)."*

2. Sections 2.1 and 2.2: There are no aerosol data listed except for the CPC in ACE-ENA. Figure S2 also includes aircraft CCN data that should be mentioned here. More broadly, I'm surprised that the authors don't take advantage of the additional aerosol measurements available at ENA. You mention repeatedly that the aerosol concentrations at ENA better resemble the "clean" experiment than the control values, and show this for one case in Fig. S2, but it would be easy to show the issue persists during all cases and better quantify the general bias, differences in aerosol/CCN definitions notwithstanding.

\>\> Thank you for pointing out this omission. We have added the following text describing the CCN data that we use for comparison with WRF in Section 2.2, **L111-117:** *The cloud condensation nuclei concentration is obtained from the CCN-200 particle counter aboard the G-1 aircraft providing CCN at approximately 0.2% supersaturation every second (i.e., $N\_CCN\_1$ as discussed in Uin and Mei, 2019). For the comparison of the aerosol properties in clear-sky conditions with the WRF model we select only those aircraft samples within a 1∘ × 1∘ region from the ARM site and below 2 km outside clouds as determined by measured cloud water content.*

Please see below the aerosol concentration comparison with WRF from all case study flights. It is evident from this plot that the lower condensation particle concentrations (CPC) in cloud-free air sampled by the aircraft suggest that the control simulation of NWFA (number of water friendly aerosols) may generally be more polluted than the observations on most days across both seasons. Note that this is not an apples-to-apples comparison since NWFA is a bulk aerosol particle number based on a single mode log-normal size distribution derived from GOCART sulfate, organic carbon, and sea salt masses assumed to be internally mixed with a hygroscopicity factor of 0.4 and aerosol mean radius of 40 nm (Thompson and Eidhammer, 2014). Therefore, it may be more comparable to a total aerosol particle count from the CPC though accumulation mode aerosol number better characterized by CCN provides useful context. While this comparison reveals a general bias in the control run, we believe this topic merits further investigation outside of this study since our results focus more on cloud responses to changes in aerosol and simulations cover the range of CPC values observed across cases. However, for completeness, we have included this plot in the supplement.

Reference
Thompson, G., and T. Eidhammer, 2014: A Study of Aerosol Impacts on Clouds and Precipitation Development in a Large Winter Cyclone. *J. Atmos. Sci.*, 71, 3636–3658, https://doi.org/10.1175/JAS-D-13-0305.1.

[Figure]

*Figure S2. Vertical profile of the background number of water friendly aerosol (NWFA) concentrations for pristine (N1), clean (N2), control (N3), and polluted (N4) conditions for all case study days at 13:00 UTC using the Thompson Aerosol-Aware scheme plotted over the boundary layer with observations of the total condensation particle counter (CPC; black asterisks) and CCN at 0.2% supersaturation (gray asterisks) from aircraft measurements taken between 10:00 – 16:00 UTC. Note, aerosol data is omitted when total cloud water content as measured by the aircraft in the WCM-2000 data set.*

3. Lines 98-99: I don't understand how excluding this data ensures the "results remain sensitive to variation in aerosol concentration."

>> The data we are excluding are merely fixed effective radius values when LWP retrievals are not carried out due to missing microwave data. To be more precise we have replaced "results remain sensitive to variations in aerosol concentration" with "**L98-100: *However, if this information is not available we exclude it (occurring less than 30% of cases) from the analysis to avoid using fixed effective radius replacement values of 8 μm in the ARM product.***"

4. Line 108: For the MODIS retrievals shown, which channel is used? I'm assuming the default 2.1 μm? Is there any large sensitivity to this choice?

>> The other referee asked a similar question; we are pasting our response here as well. We have now provided more details regarding the effective radius retrieval in the caption and main body of the text. Specifically, we clarify that the effective radius used in the comparison is ***retrieved at 3.7-μm*** and that liquid water path and droplet number concentration are computed from effective radius and optical thickness (retrieved at 3.7-μm) in the caption. Furthermore, we add the following to section 2.3, **L123-126: *Of the three spectral channels used for $R_e$ retrievals, the sensitivity of the 3.7-μm channel is weighted closest to the cloud top, primarily due to the relatively strong absorption of water vapor at this wavelength (Platnick 2000). Because errors in the adiabatic droplet number concentrations using the 3.7-μm channel are considerably smaller than in the other bands (Grosvenor et al. 2018), we choose to use it for this study.***

5. Line 172: The issue isn't just this day, but rather a general bias throughout both seasons, correct?
>> Yes, the bias persists through both seasons. See previous comment response.

6. Lines 205-206: Why was this flight chosen as the main case study?
>> It was chosen "**L220-221: *due to the distinct closed cell features and persistence of the stratocumulus cloud deck throughout the day***" which we have added to the text.

7. Section 4.1: Why is only the case of 7/18/2017 discussed here? I understand wanting the highlight the results with one flight for illustrative purposes, but from the later figures you have results for all of the flights. It would be helpful to establish here that the case isn't an outlier and that the results are robust across the simulated days.
>> This is not an outlier case. As the text is quite long, so we chose to show our method for one of the cases (i.e. 7/18/2017) for illustrative purposes. Please see drizzle case 7/15/2017, where **L265-266: *Similar behavior is found on 7/15/2017 (as depicted in Figure S5) and generally across all case studies (discussed in section 4.3).*** Since this case is exceedingly pronounced, we show the remaining case study days in Figure R4 (below) which demonstrate the robustness of the cloud segmentation algorithm across a wide range of conditions.

[Figure]

*Figure S5. Time-series of the average (a) cloud object area, (b) minimum distance between cloud centroids, (c) minimum distance between cloud edges over each 15-minute time-interval detected for ultra clean (blue), clean (orange), control (green), and polluted (red) experiments in the case study occurring on 07/15/2017. MODIS averages (star) and standard deviations (vertical lines) are displayed on the image. LWP at 13 UTC is displayed for the clean (d) and polluted experiments (e).*

[Figure]

Figure R4. WRF simulated LWP at 13 UTC is displayed for the clean (d) and polluted experiments (e) on the remaining 8 case study days (7/6/17, 6/30/17, 6/12/17, 1/19/18, 1/24/18, 1/25/18, 1/29/18, 2/1/18).

8. Lines 231-232: I'm having trouble understanding why larger LWP differences between neighboring pixels would justify merging the objects.

>> Apologies for the confusion. The word "larger" is a typo in this sentence and we have changed it to the word **L247: *smaller.*** Note, there is a parameter in the algorithm called merg_shrd. It is a threshold to determine if two adjacent objects from the watershed segmentation should be merged or not. We first calculate the edge weight (in our case we use LWP) along the common boundary of the two objects. If the weight is smaller than "merg_thrd", then the two objects are merged into one new object. We have clarified this point in the manuscript.

9. Lines 234-235: Why use the minimum distance instead of the mean or median?

>> The minimum distance is selected since we are comparing the centroid location of one object to *all* of the other object centroid locations. If we were to use the mean or median to all other objects then there would be numerous pairs with distances that are too large to be representative of the nearest neighboring cells. The minimum distance sufficiently removes outliers. Other more complex approaches such as kd-tree distributions are outside the scope of this work.

10. Line 334/Text S2: The transfer function accounts for transmissivity (reflection and absorption), not just reflection.

>> Thanks, we have added **L351: *transmissivity (reflection and absorption)*** to this statement.

11. Lines 335-336/Eq 1/Text S2: Since you're already accounting for the clear-sky atmospheric transmissivity, this should be the surface albedo.

>> Great catch! alpha_clr was changed to ***alpha_sfc*** in these locations.

12. Lines 356-363/Table 3: I'm confused about which experiments are being used to calculate the radiative effects. Is it control-clean, or polluted-pristine? I'd imagine the absolute values should differ quite a bit between those (or other) combinations.

>> We have made several changes to the manuscript to clarify the method used to quantify the aerosol indirect effect. First, we have added, **L356-358: *There are six possible pairs which include, polluted − control Δ(N4−N3), polluted − clean Δ(N4−N3), polluted − pristine Δ(N4−N1), control − clean Δ(N3−N2), control − pristine Δ(N3−N1), and clean − pristine Δ(N2−N1).*** The cloud properties and radiative effects associated with each case study are listed in ***Tables S1-S10***.

You are correct that the absolute values can change between combinations, with stronger indirect effects found between the polluted and pristine cases compared to the clean and pristine. These estimates are now fully provided in the supplementary file. We also discuss in the manuscript, **L360-361: *By using a wide range of aerosol concentrations we aim to capture variability in ACI but acknowledge that non-linearity in the relationship between cloud variables with $N_d$ may be missed from the use of only 4 aerosol experiments.***

Note, during the process of adding the additional tables we identified a bug in the radiative effect calculation. The first version used the daily max incoming solar radiation instead of the dailymean. This resulted in radiative effects that were overly large. These numbers are now also included in the tables to clear up any confusion. Overall, this change did not impact the significance or general trend of the results (since they were all scaled by the same bias).

13. Line 370: Aren't the glaciation effects in Christensen et al. (2014) thought to arise from INP, not just CCN? Are there any INP differences in the experiments?
>> Good point! As we are not perturbing the number of "ice-friendly" nuclei within the Thompson Aerosol-Aware scheme, we do not expect similar glaciation indirect effects as observed in ship tracks by Christensen et al. (2014). We have removed this reference and revised the lines to the following, **L389-395:** *Although the Thompson microphysics scheme considers ice multiplication from rime-splinters through the Hallett–Mossop process (Hallet and Mossop, 1974) , a phenomenon known to lead to cloud morphology breakup and alteration, accompanied by enhanced precipitation (Abel et al. 2017; Eirund et al. 2019).*

Added references
Abel, S. J., Boutle, I. A., Waite, K., Fox, S., Brown, P. R. A., Cotton, R., Lloyd, G., Choularton, T. W., and Bower, K. N.: The Role of Precipitation in Controlling the Transition from Stratocumulus to Cumulus Clouds in a Northern Hemisphere Cold-Air Outbreak, Journal of the Atmospheric Sciences, 74, 2293 – 2314, https://doi.org/https://doi.org/10.1175/JAS-D-16-0362.1, 2017.

Eirund, G. K., Lohmann, U., and Possner, A.: Cloud Ice Processes Enhance Spatial Scales of Organization in Arctic Stratocumulus, Geophysical Research Letters, 46, 14,109–14,117, https://doi.org/https://doi.org/10.1029/2019GL084959, 2019.

Hallett, J. and Mossop, S.: Production of secondary ice particles during the riming process, Nature, 249, 26–28, https://doi.org/https://doi.org/10.1038/249026a0, 1974.

14. Lines 373-374: Is this boilerplate, or do you mean it? The IPCC is fairly happy to ignore ice-phase aerosol effects as likely small, albeit highly uncertain. Do your results suggest that's a mistake? (I don't really see that from the paper, but am open to the argument more generally.)
>> We have toned down the claims on aerosol impacts on ice phase clouds since this is not a key part of the research. The lines have been revised to read as follows**, L389-395:** *Figure S12 reveals the presence of ice on 1/24/18 and 1/25/18, and intriguingly, the Twomey effect and rapid adjustments exhibit comparable agreement in these cases, as seen in the warm cloud case study days (Figure 10).* And also, *we haven't altered ice-friendly nuclei concentrations in this study. Modifying such concentrations could offer additional insights into aerosol-ice cloud interactions in future research.*

15. Section 4.3.2: The decision to neglect the cloud fraction adjustments should be given higher real estate here as a caveat, especially as the Morrison microphysics doesn't allow for full positive aerosol-cloud-precipitation feedback cycle as simulated in some LES (e.g., Yamaguchi et al. 2017). This could have a dramatic influence on cloud fraction (e.g., Goren et al. 2019, Diamond et al. 2022).
>> As running the Morrison microphysics code with fixed droplet concentration is not a primary part of this work it was given less scrutiny here, but we agree that more caveats should be discussed when using fixed $N_d$ experiments. Therefore, we have added the following statements to section 4.3.2, **L406-410:** *Note, running the Morrison microphysics scheme with fixed droplet number concentration does not allow for a full positive aerosol-cloud-precipitation feedback cycle as simulated in some LES simulations (e.g., Yamaguchi et al. 2017). This has*

*been shown to have a significant influence on the mesoscale structure of clouds, and hence, cloud fraction (e.g., Goren et al. 2019, Diamond et al. 2022), potentially having a significant impact on the net radiative effect in this sensitivity study.*"

Yamaguchi, T., Feingold, G., and Kazil, J.: Stratocumulus to Cumulus Transition by Drizzle, Journal of Advances in Modeling Earth Systems, 9, 2333-2349, 10.1002/2017MS001104, 2017.
Goren, T., Kazil, J., Hoffmann, F., Yamaguchi, T., and Feingold, G.: Anthropogenic Air Pollution Delays Marine Stratocumulus Break-up to Open-Cells, Geophysical Research Letters, 46, 14135–14144, 10.1029/2019gl085412, 2019.
Diamond, M. S., Saide, P. E., Zuidema, P., Ackerman, A. S., Doherty, S. J., Fridlind, A. M., Gordon, H., Howes, C., Kazil, J., Yamaguchi, T., Zhang, J., Feingold, G., and Wood, R.: Cloud adjustments from large-scale smoke–circulation interactions strongly modulate the southeastern Atlantic stratocumulus-to-cumulus transition, Atmos. Chem. Phys., 22, 12113-12151, 10.5194/acp-22-12113-2022, 2022.
>> Thank you for the references.

16. Line 453: The "prior observations" phrasing is misleading, as the adjustments in this work are not observational.
>> Good catch, the phrasing was misleading. Indeed, we did not use observations to compute aerosol radiative effects. The words "in prior" have been replaced by "satellite."

17. Lines 462-463: The phrasing here is a bit awkward, as it reads like autoconversion, not the underestimate of autoconversion, delays raindrop formation.
>> We have re-phrased this sentence for clarity. **L486-489: *Although, the absence of a negative LWP response in our study may be attributed to a variety of processes. First, uncertainties in the autoconversion rate (a tunable parameter that affects the formation rate of raindrops) may lead to a positive LWP response as droplet number concentrations increase if this rate is underestimated (Mülmenstädt et al., 2020; Christensen et al., 2023).***

Added references
Christensen, M. W., Ma, P.-L., Wu, P., Varble, A. C., Mülmenstädt, J., and Fast, J. D.: Evaluation of aerosol–cloud interactions in E3SM using a Lagrangian framework, Atmospheric Chemistry and Physics, 23, 2789–2812, https://doi.org/10.5194/acp-23-2789-2023, 2023.

Mülmenstädt, J., Nam, C., Salzmann, M., Kretzschmar, J., L'Ecuyer, T. S., Lohmann, U., Ma, P.-L., Myhre, G., Neubauer, D., Stier, P., Suzuki, K., Wang, M., and Quaas, J.: Reducing the aerosol forcing uncertainty using observational constraints on warm rain processes, Science Advances, 6, eaaz6433, https://doi.org/10.1126/sciadv.aaz6433, 2020.

18. Line 471: Do any of the simulations show aerosols closing open cells into closed cells? I don't think any of the figures shows this clearly. Should it be more like "aerosols expand the width of closed cells"?
>> You are correct. We do not explicitly simulate the "closing of open cells." The clouds also do not always take on the classical shape of "open" or "closed" cells so we further generalize to "stratocumulus cells" to avoid confusion. We have taken your suggestion and changed the wording here and throughout where closed and open cells are being referenced (e.g., **L501: *expand the area of stratocumulus cells***).

19. Lines 481-483: HX isn't mentioned in the author contributions.

>> Heng Xiao's contribution has been added to the revised paper.

20. Figure 4: I assume the white stars are the object centroids? This should be mentioned in the caption.
>> Yes, the white stars are object centroid locations. A legend and description has been added to the figure.

21. Figure 9: Why not just show TKE in panel h?
>> QKE is the standard output from WRF representing 2*TKE. As this quantity is not typically used across the literature, I have converted it to TKE and modified the caption accordingly.

---

## Author Response (AR3)

Response to Reviewers and Editor

*Comments in black*
Manuscript changes in **bold blue** at the line numbers of the revised draft.

Dear Dr. Christensen,

Thank you for the revised manuscript. The reviewers are generally satisfied with your document, however Reviewer 1 continues to be only marginally convinced by the ability of the model to resolve entrainment effects at cloud top. They also note some peculiarities in the physics of Figure 9 that I would agree are a bit difficult to understand. I suggest reconsideration of those profiles, but also at the very least add some further discussion of any prior studies that may help bolster an argument that a coarse resolution model is sufficient for application to aerosol cloud interactions in stratiform clouds with open cells.

Best regards,

Tim Garrett

Report #1
The authors have done a nice job responding to the reviewer comments. I recommend acceptance as-is. -MD

Report #2
I appreciate that the authors have addressed my previous comments comprehensively. I think the manuscript has significantly improved overall. The authors explained the important limitations about the model to minimize the confusion, but I am not sure my biggest concern was adequately addressed. The magnitude of LWP and cloud cover effects looks very large, and the LW heating rate does not match TKE and W' (Fig.9b, e, and d). Is it possible to say that the dynamic effects of cloud-top radiative cooling are adequately resolved in WRF? It would be nice to add some explanations and references if possible.

Dear Tim Garrett,

We appreciate the insightful feedback provided by Reviewer 2 and the opportunity to further refine our manuscript based on these comments. We understand the concerns raised by the reviewer and have made several key changes to Figure 9 to clear up confusion. First, we fixed a typo. The vertical velocity variance was inaccurately labeled as w' when it should have been $w'^2$. Second, we have added a shaded layer to each panel to represent the portion of the vertical profile with cloud so that it is easier to distinguish the location of the cloud top from the cloud base. Third, we now provide the TKE based on the subgrid parameterized output from MYNN3 as well as the resolved 3D wind field to account for turbulence by convective eddies at 800-m grid-spacing. The TKE computed based on the resolved winds show relatively larger values in the vicinity of the cloud top. Based on this feature as well as the evidence presented in Figure S10 and Figure R1 from our first round of revisions we are confident that turbulent kinetic energy production in the planetary boundary layer is connected to the cloud and radiation schemes. However, we agree with the reviewer that stronger liquid water path and cloud fraction adjustments to changes in aerosol concentration may arise if the parameterized entrainment rates are too weak in the MYNN scheme at relatively coarse resolution compared to LES. Therefore, we have tempered our language regarding these strong associations in the manuscript. We have also meticulously described our model configuration and setup, along with a thorough discussion of the model's strengths and limitations, to facilitate reproducibility and expansion of this analysis in future work.

Please see below the key points raised by the reviewer and the changes we have made to the manuscript.

Best regards,

Matthew Christensen

Editor Comment 1
*Any prior studies that may help bolster an argument that a coarse resolution model is sufficient for application to aerosol cloud interactions in stratiform clouds with open cells.*
>>The following has been added to the conclusions section:

**Lines 487 – 497: As computation power increases, km-scale models employed with PBL schemes (similar to ours) will increasingly be used to quantify aerosol-cloud interactions at global-scales with increasing complexity (Terai et al. 2020). Kilometer scale models have been shown to successfully simulate the properties and mesoscale structure of stratocumulus. Chen et al. (2022) used WRF with 1-km grid spacing to simulate the roll structure and transition of stratocumulus and cloud streets by gradients in sea surface temperature. Saffin et al. (2023) utilize the Met Office Unified Model to simulate cloud transitions observed during the ATOMIC field campaign at similar scales. This transition shows the development of small shallow clouds into larger flower-type clouds with detrainment, triggered by increased mesoscale organization over several tens of kilometers. Beucher et al. (2022) utilized the French convection-permitting model AROME-OM at kilometer scales, successfully simulating four primary mesoscale patterns observed during the EUREC4A campaign. Despite the success of simulating the realism of the mesoscale structure of marine stratocumulus, further refinement may be needed to enhance connections between radiation, microphysics, and planetary boundary layer schemes for adequately simulating the complexity of aerosol-cloud interactions.**

Reviewer Comment 1
*Is it possible to say that the dynamic effects of cloud-top radiative cooling are adequately resolved in WRF? It would be nice to add some explanations and references if possible.*
>> First, we have added a panel to Figure 9 which shows the resolved TKE. This quantity is described in section 4.2.2.

**L322 – L329: To account for the turbulence of the convective eddies at 800-m grid spacing (in the so-called "gray-zone" where eddies in the PBL are partially resolved; Shin and Dudhia, 2016), TKE is also provided using the 3D resolved winds (Figure 9i) following the equation: $TKE = \frac{1}{2}(u'^2 + v'^2 + w'^2)$, where u'², v'² and w'² are the variances of the winds computed over 3.2 x 3.2 km² regions. The resolved TKE is not very sensitive to changes in the averaging scale in which the 3.2, 6.4, and 12.8 km scales show similar magnitude within the cloud layer. While the TKE computed using the resolved winds shows a relative increase near cloud top hinting at a better connection to the cloud top radiative flux profile compared to the subgrid TKE output from MYNN3, this is a relatively weak relationship compared to large eddy simulations of stratocumulus (McMichael et al., 2019), and may suggest further refinement is needed in connecting these processes within the MYNN Eddy-Diffusivity Mass Flux (EDMF) scheme (Olson et al. 2019). Possible implications of the relatively weak mixing on the liquid water path and cloud fraction adjustments are discussed in further detail in the conclusions section.**

Previous version Figure 9

[Figure]

**Figure 9.** Vertical profile of the a) mean shortwave and b) longwave radiative heating rate, c) mean vertical velocity, d) mean horizontal wind speed, e) vertical velocity variance, and h) turbulent kinetic energy (TKE) for pristine (blue), unpolluted (orange), control (green), and polluted (red) WRF simulations on 07/15/2017 at 13:00 UTC.

Revised Figure 9

[Figure]

>> Next, the following has been added to the conclusions section:

**Lines 516 – Lines 533: Ghonima et al. (2017) evaluated the MYNN scheme and other turbulence parameterization schemes using single-column model experiments showing that entrainment flux tendencies in stratocumulus tend to be underestimated compared to LES, resulting in cooler, moister stratocumulus-topped boundary layers. This discrepancy may imply a deficiency in representing strong turbulent mixing near the cloud top in our simulations. However, our simulations show an enhanced peak in the resolved TKE near the top of the stratocumulus cloud layer (Figure 9i). Also, when radiation is deactivated, TKE is much smaller and the cloud layer becomes significantly shallower (Figure S10), highlighting the role of radiative processes in driving stronger TKE throughout the boundary layer. WRF version 4.2 introduced scale-awareness, dynamically adjusting parameterized turbulent kinetic energy as resolution decreases, thus offering a more explicit representation of turbulent processes at finer scales (Olson et al. 2019). Subgrid-scale clouds produced by the MYNN-EDMF (section 3) are coupled to the longwave and shortwave radiation schemes (namelist parameter icloud_bl is set to 1). Despite these couplings, uncertainties may persist due to relatively coarse vertical resolution (compared to LES) and the ability to capture nonlocal production of TKE associated with cloud-top radiative cooling. Alternative approaches, such as explicit entrainment or employing the mass-flux method for downdrafts, may offer improved parameterization of destabilized parcels in stratocumulus environments (Olson et al. 2019). The impacts of model caveats**

**like these on cloud cell expansion due to increased aerosol concentration should be explored in subsequent research with higher resolution models including LES where the cloud-top entrainment interface can be modeled at finer spatial scale resolutions. Nevertheless, our model set up shows evidence that radiative cooling drives stronger turbulence in the marine boundary layer but it remains crucial to constrain such parameters based on observations (Suzuki and Stephens, 2009; Golaz et al. 2013; Christensen et al., 2023; Varble et al., 2023), where possible, to enhance model development and our understanding of aerosol-cloud interactions and radiative forcing.**

Reviewers Comment 2

*The magnitude of LWP and cloud cover effects looks very large.*

>> The following have been added to the conclusions: Previous studies, such as Gryspeerdt et al. (2020) and Bellouin et al. (2020), have reported enhancement factors for radiative forcing attributable to aerosol-cloud interactions, when combined, reaching as high as 150% for adjustments in liquid and cloud fraction. **L392 – 396: Consequently, our findings approach the upper limits of these adjustments possibly due to a weak connection between entrainment mixing and cloud top radiation from the use of km-scale models (discussed further in the conclusions).**

Reviewers Comment 3

*The LW heating rate does not match TKE and W' (Fig.9b, e, and d)*

>> We agree, the TKE profile output from the MYNN scheme at the subgrid scale resolution *does not match* the 'spike' in the radiative cooling rates near the cloud top. In addition to the subgrid scale TKE provided by the MYNN scheme, the resolved TKE is now also computed from the grid-scale wind variances. This data has been added to Figure 9. At the typical gray-zone spatial resolutions (~1 km grid-spacings), the convective eddies are partly resolved (shown via our filtered resolved TKE) and partly parameterized (shown via the MYNN SGS TKE). The resolved TKE is part of the response to cloud-top radiative cooling which we observe as a maximum near the cloud tops. Hence, the relative increase in TKE near the cloud top indicates some coupling between the longwave heating rate and the TKE, although, as stated above (and in the manuscript) this connection may be relatively weak compared to finer-scale LES models.

Finally, it is noteworthy pointing out that in versions of WRF before V4.2, it was reported in Puhales et al. (2023) that the TKE budget terms were unbalanced. However, this artifact only affected the diagnostic output history files, not the actual TKE budget. Upon investigating this we discovered a typo in our original manuscript which we referenced WRF V4.2; we are actually using WRF V4.3. Nevertheless, the cited discrepancy in the WRF version is not relevant to the changes in the MYNN scheme which took place in prior versions of the WRF model.

References

Bellouin, N., Quaas, J., Gryspeerdt, E., Kinne, S., Stier, P., Watson-Parris, D., Boucher, O., Carslaw, K. S., Christensen, M., Daniau, A. L., Dufresne, J. L., Feingold, G., Fiedler, S., Forster, P.,

Gettelman, A., Haywood, J. M., Lohmann, U., Malavelle, F., Mauritsen, T., . . . Stevens, B. (2020). Bounding Global Aerosol Radiative Forcing of Climate Change. Reviews of Geophysics, 58(1). https://doi.org/ARTN e2019RG00066010.1029/2019RG000660

Beucher, F., Couvreux, F., Bouniol, D., Faure, G., Favot, F., Dauhut, T., & Ayet, A. (2022). Process-oriented evaluation of the oversea AROME configuration: Focus on the representation of cloud organisation. Quarterly Journal of the Royal Meteorological Society, 148(749), 3429-3447. https://doi.org/10.1002/qj.4354

Chen, J. Y., Wang, H. L., Li, X. Y., Painemal, D., Sorooshian, A., Thornhill, K. L., Robinson, C., & Shingler, T. (2022). Impact of Meteorological Factors on the Mesoscale Morphology of Cloud Streets during a Cold-Air Outbreak over the Western North Atlantic. Journal of the Atmospheric Sciences, 79(11), 2863-2879. https://doi.org/10.1175/Jas-D-22-0034.1

Christensen, M. W., Ma, P.-L., Wu, P., Varble, A. C., Mülmenstädt, J., and Fast, J. D.: Evaluation of aerosol–cloud interactions in E3SM using a Lagrangian framework, Atmospheric Chemistry and Physics, 23, 2789–2812, https://doi.org/10.5194/acp-23-2789-2023, 2023.

Golaz, J.-C., Horowitz, L. W., and Levy, H.: Cloud tuning in a coupled climate model: Impact on 20th century warming, Geophysical Research Letters, 40, 2246–2251, https://doi.org/10.1002/grl.50232, 2013.

Ghonima, M. S., Yang, H., Kim, C. K., Heus, T., & Kleissl, J. (2017). Evaluation of WRF SCM Simulations of Stratocumulus-Topped Marine and Coastal Boundary Layers and Improvements to Turbulence and Entrainment Parameterizations. Journal of Advances in Modeling Earth Systems, 9(7), 2635-2653. https://doi.org/10.1002/2017ms001092

Gryspeerdt, E., Mülmenstädt, J., Gettelman, A., Malavelle, F. F., Morrison, H., Neubauer, D., Partridge, D. G., Stier, P., Takemura, T., Wang, H., Wang, M., and Zhang, K.: Surprising similarities in model and observational aerosol radiative forcing estimates, Atmos. Chem. Phys., 20, 613–623, https://doi.org/10.5194/acp-20-613-2020, 2020.

McMichael, L. A., Mechem, D. B., Wang, S. P., Wang, Q., Kogan, Y. L., and Teixeira, J.: Assessing the mechanisms governing the daytime evolution of marine stratocumulus using large-eddy simulation, Quarterly Journal of the Royal Meteorological Society, 145, 845–866, https://doi.org/10.1002/qj.3469, 2019.

Puhales, F. S., J. B. Olson, J. Dudhia, D. L. de Bem, R. Maroneze, O. C. Acevedo, F. D. Costa, and V. Anabor, (2023), Turbulent Kinetic Energy Budget for MYNN-EDMF PBL Scheme in WRF model, Technical note UFSM/GruMA 001/2023, Universidade Federal De Santa Maria.

Olson, J. B., J. S. Kenyon, W. A. Angevine, J. M. Brown, M. Pagowski, and K. Suselj (2019), A Description of the MYNN-EDMF Scheme and the Coupling to Other Components in WRF-ARW, NOAA Technical Memorandum, https://doi.org/10.25923/n9wm-be49.

Saffin, L., Lock, A., Tomassini, L., Blyth, A., B.ing, S., Denby, L., & Marsham, J. (2023). Kilometer-scale simulations of trade-wind cumulus capture processes of mesoscale organization. Journal of Advances in Modeling Earth Systems, 15, e2022MS003295. https://doi.org/10.1029/2022MS003295

Shin, H. H. and Dudhia, J.: Evaluation of PBL Parameterizations in WRF at Subkilometer Grid Spacings: Turbulence Statistics in the Dry Convective Boundary Layer, Monthly Weather Review, 144, 1161–1177, https://doi.org/10.1175/Mwr-D-15-0208.1, 2016.

Suzuki, K. and Stephens, G. L.: Relationship between radar reflectivity and the time scale of warm rain formation in a global cloud-resolving model, Atmos. Res., 12.010, doi:10.1016/j.atmosres., 2009.

Terai, C. R., Pritchard, M. S., Blossey, P., & Bretherton, C. S. (2020). The impact of resolving subkilometer processes on aerosol-cloud interactions of low-level clouds in global model simulations. Journal of Advances in Modeling Earth Systems, 12, e2020MS002274. https://doi.org/ 10.1029/2020MS002274

Varble, A. C., Ma, P.-L., Christensen, M. W., Mülmenstädt, J., Tang, S., and Fast, J.: Evaluation of Liquid Cloud Albedo Susceptibility in E3SM Using Coupled Eastern North Atlantic Surface and Satellite Retrievals, EGUsphere, 2023, 1–39, https://doi.org/10.5194/egusphere-2023-998, 2023.